# Aryl hydrocarbon receptor is required for optimal B-cell proliferation

Matteo Villa[1], Manolis Gialitakis[1], Mauro Tolaini[1], Helena Ahlfors[1,†], Colin J Henderson[2], C Roland Wolf[2], Robert Brink[3] & Brigitta Stockinger[1,*]

## Abstract

**The aryl hydrocarbon receptor (AhR), a transcription factor known for mediating xenobiotic toxicity, is expressed in B cells, which are known targets for environmental pollutants. However, it is unclear what the physiological functions of AhR in B cells are. We show here that expression of *Ahr* in B cells is up-regulated upon B-cell receptor (BCR) engagement and IL-4 treatment. Addition of a natural ligand of AhR, FICZ, induces AhR translocation to the nucleus and transcription of the AhR target gene *Cyp1a1*, showing that the AhR pathway is functional in B cells. AhR-deficient ($Ahr^{-/-}$) B cells proliferate less than AhR-sufficient ($Ahr^{+/+}$) cells following *in vitro* BCR stimulation and *in vivo* adoptive transfer models confirmed that $Ahr^{-/-}$ B cells are outcompeted by $Ahr^{+/+}$ cells. Transcriptome comparison of AhR-deficient and AhR-sufficient B cells identified cyclin O (*Ccno*), a direct target of AhR, as a top candidate affected by AhR deficiency.**

**Keywords** aryl hydrocarbon receptor; B cells; cyclin O; proliferation
**Subject Categories** Immunology
**The EMBO Journal (2017) 36: 116–128**

## Introduction

The aryl hydrocarbon receptor (AhR), a ligand-dependent transcription factor that responds to environmental signals, is widely expressed in the haematopoietic system. As the AhR had originally been defined as the receptor for dioxins and other chemical pollutants, the effect of those substances on the development and function of immune cells was a focus in toxicological research for many years (Kerkvliet, 2002). However, more recently the emphasis shifted towards attempts to delineate the physiological functions of AhR, which are indicated by its strong evolutionary conservation from invertebrates onwards (Hahn *et al*, 1997). These studies were facilitated by the availability of AhR-deficient mice (Fernandez-Salguero *et al*, 1995; Schmidt *et al*, 1996; Mimura *et al*, 1997).

Although such mice exhibit no overt immunological phenotype in steady state, alterations in immune responses and cell types are revealed upon immunological challenge, indicating a substantial, if currently insufficiently characterized role for AhR in many cell types of the immune system, reviewed in (Stockinger *et al*, 2014). In B cells, AhR is widely expressed in developmental subsets and upon activation (Marcus *et al*, 1998; Tanaka *et al*, 2005; Sherr & Monti, 2013), and AhR engagement by environmental toxins was shown to lead to suppression of humoral immune responses (Kerkvliet *et al*, 1990).

However, the physiological importance of AhR expression in B cells remains ill defined. Here, we investigated the impact of AhR deficiency on B-cell function in the absence of xenobiotic influences.

We generated mice with B cell-specific deletion of *Ahr* via the Cre-*loxP* system. AhR deficiency had no influence on B-cell responses to T-dependent and T-independent antigens. However, AhR-deficient B cells exhibited reduced ability to proliferate, being less prone to enter the S phase of the cell cycle. As a consequence, $Ahr^{-/-}$ B cells were unable to compete with $Ahr^{+/+}$ B cells and were impaired in their ability to reconstitute an empty host or mount an antigen-dependent proliferative response *in vivo*. Global comparison of the transcriptome of AhR-deficient and AhR-sufficient B cells identified cyclin O (*Ccno*) as one of the top candidates affected by AhR deficiency, and ChIP analysis of the *Ccno* locus showed it to be directly regulated by AhR.

## Results

### Expression of AhR in B cells is induced upon B-cell receptor activation

Aryl hydrocarbon receptor expression in B cells has been previously shown (Marcus *et al*, 1998; Tanaka *et al*, 2005), but these studies did not explore the full repertoire of B-cell activation stimuli and were largely based on using total splenocytes or cell lines. In order to define the levels of *Ahr* expression in different developmental subsets of B cells, we FACS purified B-cell subsets from bone

1   The Francis Crick Institute, Mill Hill Laboratory, London, UK
2   Division of Cancer Research, University of Dundee Ninewells Hospital and Medical School, Dundee, UK
3   Garvan Institute of Medical Research, Sydney, NSW, Australia
    *Corresponding author. Tel: +44 2037961600; E-mail: brigitta.stockinger@crick.ac.uk
    †Present address: Great Ormond Street Hospital, NE Thames Regional Genetics Service Laboratories, London, UK

   

marrow, spleen, peritoneal cavity and Peyer's patches of non-immune C57Bl/6 mice. *Ahr* was expressed across most subsets, albeit at lower levels in bone marrow Pro and PreB cells and germinal centre (GC) B cells. The highest expression was found in splenic marginal zone B cells (MZB), peritoneal CD5$^+$ B1 cells and bone marrow-resident plasma cells (PCs) (Figs 1A and EV1A). The expression levels of *Ahr* in total spleen B220$^+$ B cells were similar to that of T$_H$17 cells (Fig EV1B) and among splenic subsets MZB cells expressed the highest levels of *Ahr* (Fig EV1C). Activation of B cells through the BCR, and to some degree with IL-4, resulted in substantial up-regulation of *Ahr*, whereas TLR ligands such as LPS or CpG as well as CD40 ligand and BAFF did not affect *Ahr* levels (Fig 1B). We further explored whether BCR crosslinking and IL-4

could synergize in inducing *Ahr* expression. As shown in Fig 1C–E, co-stimulation of B cells with anti-IgM and IL-4 substantially increased AhR mRNA and protein expression as compared to the single treatments. The increase in *Ahr* expression upon BCR stimulation with anti-IgM (α-IgM) was seen across all subsets of splenic B cells (Fig 1F). AhR expression peaked after 4 h of stimulation with anti-IgM and IL-4 and steadily decreased over time approaching steady-state levels by 24 h (Fig 1G).

Regulation of *Ahr* expression had previously been linked to the canonical NF-κB pathway, albeit in mouse embryonic fibroblasts (Vogel *et al*, 2014). We assessed the potential contribution of this pathway in our system using the IKKβ inhibitor BI605906, which blocks the degradation of IκBα, thereby preventing nuclear

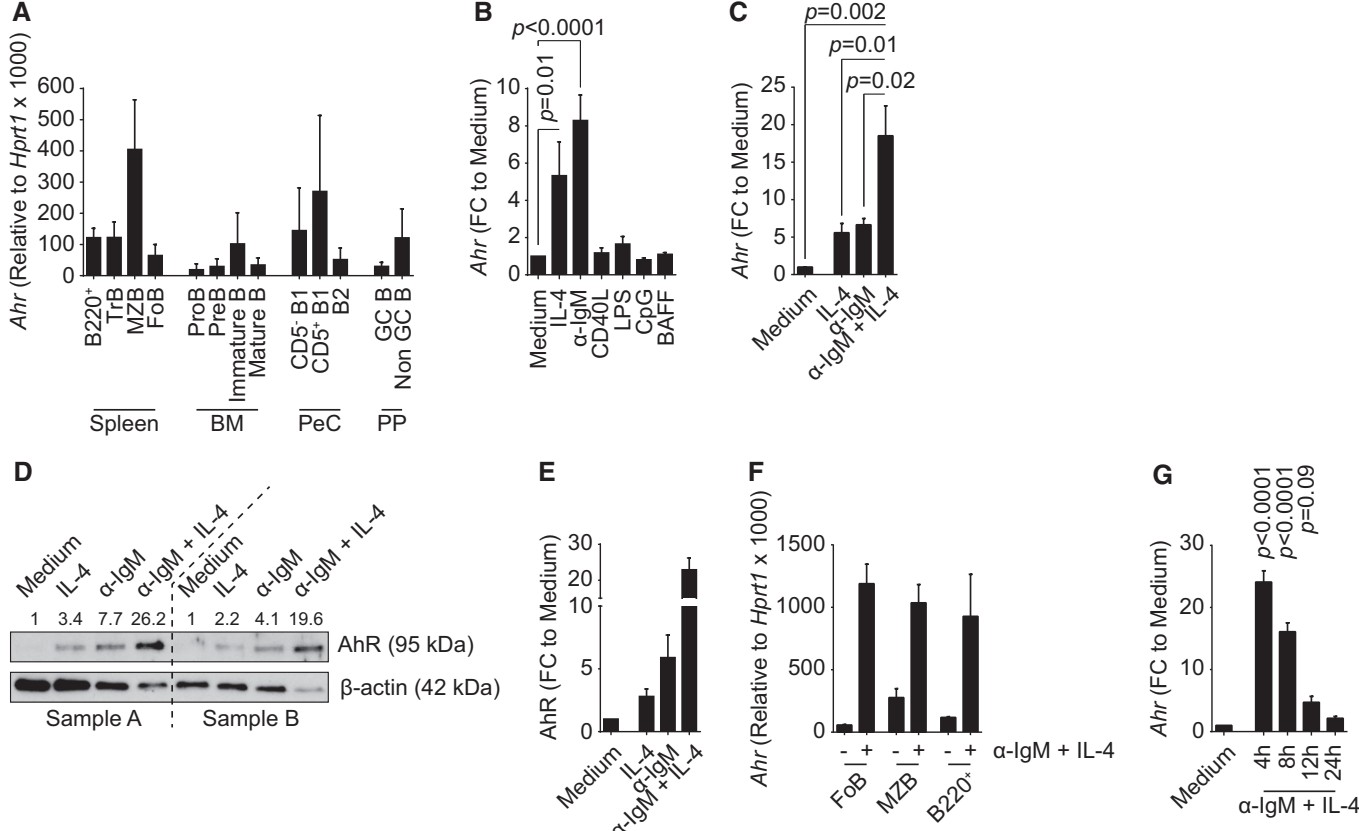

**Figure 1.  B-cell activation via BCR engagement and/or IL-4 up-regulates *Ahr* expression.**

A   qPCR analysis of *Ahr* expression in B-cell subsets purified from C57Bl/6 mice. *Ahr* expression was normalized to *Hprt1*. n = 3–5 independent experiments; mean ± SD.

B   qPCR analysis of *Ahr* expression in splenic CD19$^+$ cells isolated from C57Bl/6 mice and cultured for 4 h as indicated. *Ahr* expression was normalized to *Hprt1*. *Ahr* expression among groups was normalized to medium. n = 4 independent experiments; mean ± SEM; one-way ANOVA, Dunnett's multiple comparison test.

C   qPCR analysis of *Ahr* expression in splenic CD19$^+$ cells isolated from C57Bl/6 mice and cultured for 4 h with 20 ng/ml IL-4 and/or 10 μg/ml α-IgM. *Ahr* expression was normalized to *Hprt1*. *Ahr* expression among groups was normalized to medium. n = 3 independent experiments; mean ± SEM; one-way ANOVA, Tukey's multiple comparison test.

D   Western blot of protein extracts from splenic CD19$^+$ cells isolated from C57Bl/6 mice and cultured for 4 h with 20 ng/ml IL-4 and/or 10 μg/ml α-IgM. Values above the blots indicate AhR protein quantification obtained by densitometry, normalized to β-actin and compared to medium sample. Samples A and B indicate two independent replicates.

E   Quantification by densitometric analysis of the results shown in (D). n = 2 independent experiments; mean ± range.

F   qPCR analysis of *Ahr* expression in purified splenic B-cell subsets isolated from C57Bl/6 mice and cultured as indicated for 4 h. *Ahr* expression was normalized to *Hprt1*. n = 2 independent experiments; mean ± range.

G   qPCR analysis of *Ahr* expression in splenic CD19$^+$ cells isolated from C57Bl/6 mice and cultured for the indicated time points with 20 ng/ml IL-4 and/or 10 μg/ml α-IgM. *Ahr* expression was normalized to *Hprt1*. *Ahr* expression among groups was normalized to medium. n = 5 independent experiments; mean ± SEM; one-way ANOVA, Dunnett's multiple comparison test.

translocation of NF-κB. However, inhibition of NF-κB did not influence *Ahr* up-regulation upon BCR stimulation (Fig EV1D–F). AhR is therefore expressed in steady-state B cell and further induced upon engagement of the BCR in an NF-κB-independent fashion.

### Nuclear translocation and activation of AhR in B cells

We next determined the translocation of AhR from its cytoplasmic localization to the nucleus following exposure to ligand. Western blot analysis of cytoplasmic and nuclear fractions of α-IgM-activated B cells exposed to either the vehicle control DMSO, the high-affinity endogenous ligand FICZ or the AhR inhibitor CH223191 showed increased nuclear translocation upon exposure to FICZ, although there was some nuclear AhR detectable in the control samples too (Fig 2A). This could be due to the presence of tryptophan in culture medium that is rapidly metabolized to form the AhR ligand FICZ (Veldhoen *et al*, 2009); however, we cannot exclude also a direct

effect of DMSO in driving AhR translocation into the nucleus. In contrast, the presence of the AhR inhibitor reduced nuclear translocation, also compared to the DMSO control.

As a consequence of nuclear translocation following AhR activation, the downstream target *Cyp1a1* was induced (Fig 2B). This required both activation and exposure to AhR agonist and was restricted to activation via the B-cell receptor. Although IL-4 treatment of B cells increased their expression of *Ahr*, IL-4 in conjunction with FICZ did not activate the AhR pathway and therefore did not result in induction of *Cyp1a1*. In order to investigate AhR activation in B cells on the single cell level, we took advantage of a mouse model in which AhR activation can be traced by expression of a reporter gene. This knock-in mouse strain bears a gene encoding Cre recombinase under control of the endogenous *Cyp1a1* promoter (Henderson *et al*, 2015). Breeding these mice with reporter mice expressing eYFP from the *Rosa26* promoter allowed visualizing cells that had activated the AhR pathway via eYFP expression.

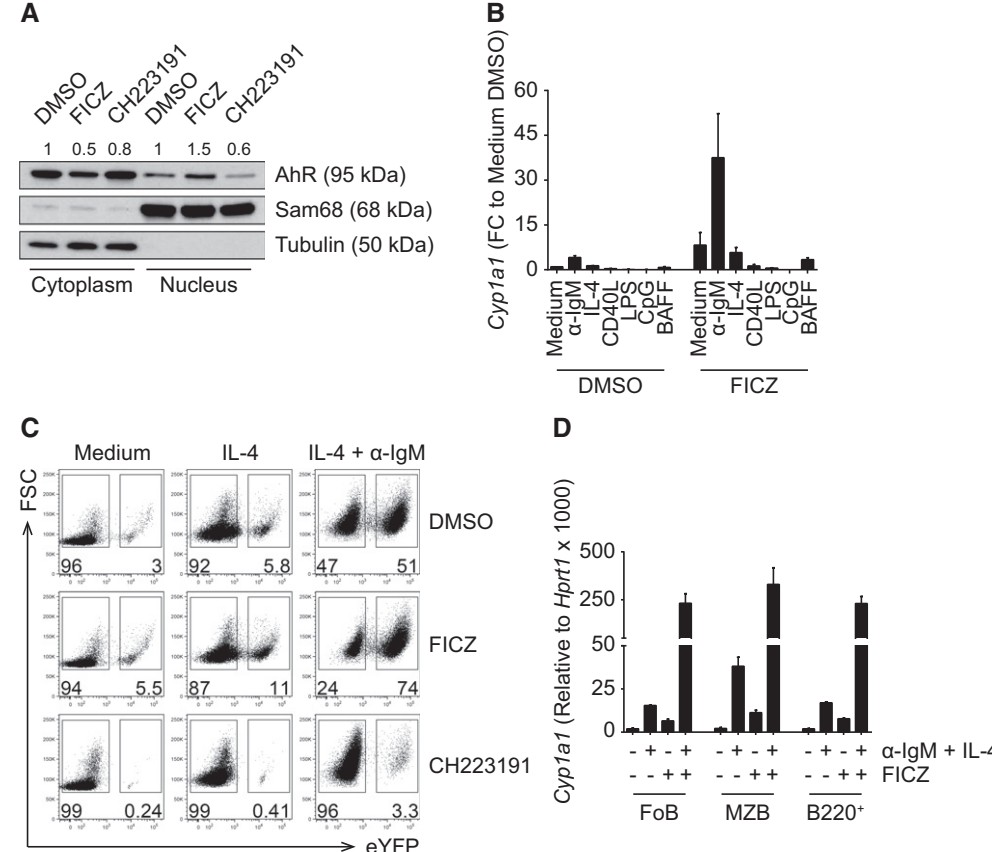

**Figure 2.  AhR translocates to the nucleus and induces *Cyp1a1* expression upon BCR engagement and in the presence of FICZ *in vitro*.**

A  Western blot of nuclear and cytoplasmic protein extracts from splenic CD19+ cells isolated from C57Bl/6 mice and cultured with anti-IgM (α-IgM) for 4 h, followed by the indicated treatment for 2 h. Values above the blots indicate AhR protein quantification obtained by densitometry, normalized to Sam68 or tubulin and compared to the DMSO-treated sample. Representative data of *n* = 3 independent experiments.

B  qPCR analysis of *Cyp1a1* expression in splenic CD19+ cells isolated from C57Bl/6 mice and cultured for 24 h as indicated. *Cyp1a1* expression was normalized to *Hprt1*. *Cyp1a1* expression among groups was normalized to medium DMSO. *n* = 2 independent experiments; mean ± range.

C  Flow cytometry analysis of Cyp1a1 expression, reported by eYFP, in splenic CD19+ cells isolated from *Cyp1a1^{Cre} R26R eYFP* mice and cultured for 72 h as indicated. Representative data of *n* = 3 independent experiments.

D  qPCR analysis of *Cyp1a1* expression in purified splenic B-cell subsets isolated from C57Bl/6 mice and cultured as indicated for 24 h. *Cyp1a1* expression was normalized to *Hprt1*. *n* = 2 independent experiments; mean ± range.

As shown in Fig 2C, B cells from $Cyp1a1^{Cre}$ reporter mice, cultured either without stimulation (medium), with IL-4 or with the combination of IL-4 and α-IgM showed increased eYFP expression upon BCR stimulation, already under baseline conditions without addition of AhR agonist. Addition of FICZ markedly increased eYFP expression in α-IgM-stimulated cells, whereas addition of the AhR antagonist CH223191 reduced the background levels of eYFP (which are due to AhR agonists in the medium) and also suppressed eYFP expression in stimulated B cells. Optimal induction of $Cyp1a1$ expression was dependent on both B-cell receptor triggering and presence of AhR ligand and was observed across all mature B-cell subsets in the spleen (Fig 2D).

Thus, mature B cells in peripheral lymphoid organs express AhR and respond to AhR ligands by AhR translocation to the nucleus and activation of the classical AhR pathway that results in induction of $Cyp1a1$.

## Activation of antigen-specific B cells induces the AhR pathway *in vivo*

In order to investigate whether AhR is functionally active in B cells *in vivo*, we crossed the $Cyp1a1^{Cre}$ reporter mice to transgenic mice expressing a hen egg lysozyme (HEL)-specific BCR ($SW_{HEL}$ mice) (Phan *et al*, 2003). In $SW_{HEL}$ mice, 40–60% of B cells respond to HEL and these B cells are able to undergo class switching. We transferred total splenocytes from CD45.1 allotype-marked $SW_{HEL}$ $Cyp1a1^{Cre}$ mice together with sheep red blood cells (SRBCs) coupled with HEL or mock-coupled into C57Bl/6 recipients to induce a T cell-dependent response of HEL-specific B cells. In addition, some of the mice received injection of the xenobiotic AhR ligand 3-methyl-cholanthrene (3-MC), since FICZ is rapidly metabolized *in vivo* (Fig 3A). Seven days later, eYFP expression in GC and non-GC B cells isolated from the spleens of recipient mice was analysed. Upon immunization with SRBC-HEL, the transferred B cells expanded and formed GC (Fig 3B middle and lower rows), whereas mice treated with SRBC-mock had no detectable donor-derived GC B cells and even in the presence of 3-MC failed to show induction of $Cyp1a1$ (measured as eYFP, Fig 3B top row). Mice that received SRBC-HEL together with vehicle showed GC B-cell expansion, but no up-regulation of $Cyp1a1$. Only the combination of antigen-dependent activation and AhR agonist was able to drive $Cyp1a1$ expression in both GC and non-GC B cells (Fig 3B bottom row, Fig 3C and D).

This suggests that antigen stimulation of BCR, by inducing AhR expression, enhances the sensitivity of AhR pathway to available ligands and it is in line with the absence of eYFP expression in untreated $Cyp1a1^{Cre}$ reporter mice.

## AhR deficiency impairs B-cell proliferation *in vitro* and *in vivo*

In order to evaluate the consequences of AhR deficiency in B cells, we established mice that specifically lack AhR in B cells by crossing $Ahr^{+/+}$ or $Ahr^{-/-}$ $mb1^{Cre}$ mice with $Ahr^{fl/fl}$ mice to generate $Ahr^{fl/+}$ $mb1^{Cre+}$ controls and $Ahr^{fl/-}$ $mb1^{Cre+}$ offspring (Fig EV2A). $mb1^{Cre}$ mice carry a construct for Cre recombinase under the control of the B cell-specific $mb1$ promoter ($mb1$ encodes for Igα/CD79A, an essential signalling component of the BCR complex; Hobeika *et al*, 2006). The mice we generated also express a Cre-dependent reporter that marks cells as eYFP$^+$ upon Cre recombinase activation

(Fig EV2A). Cre-induced deletion of $Ahr$, as well as eYFP fate-reporting, was efficient and specific, from the earliest stages of B-cell development in the bone marrow (c-kit$^{int}$ B220$^+$ cells) to the peripheral compartment (Fig EV2B–E). We assessed the composition of the B-cell compartment and serum immunoglobulin levels at steady state in $Ahr^{fl/-}$ $mb1^{Cre+}$ and $Ahr^{fl/+}$ $mb1^{Cre+}$ non-immune mice and compared them to mice with complete deletion of AhR ($Ahr^{-/-}$). B cell-specific AhR-deficient mice and full $Ahr^{-/-}$ mice showed no major differences in the distribution of B-cell subsets (Fig EV3A and Appendix Fig S1A). However, full $Ahr^{-/-}$ mice had altered serum immunoglobulin levels as compared to $Ahr^{+/+}$ controls, showing elevated IgM but reduced IgG1 isotypes (Appendix Fig S1B and C), whereas other isotypes were expressed to similar levels (Appendix Fig S1D–G). In contrast, there were no differences in the serum immunoglobulin levels between $Ahr^{fl/-}$ $mb1^{Cre+}$ and $Ahr^{fl/+}$ $mb1^{Cre+}$ mice (Fig EV3B–G). Responses to immunization with T-independent (TNP-Ficoll) and T-dependent (NP-CGG and cholera toxin—Ctx) antigens resulted in similar antibody responses in mice with AhR-deficient or AhR-sufficient B cells (Fig EV4).

However, it was noticeable that AhR-deficient B cells (Fig 4A, black line) were compromised in their proliferative potential following BCR stimulation, compared with AhR-sufficient B cells (Fig 4A, solid grey). B cells from heterozygous $Ahr^{+/-}$ mice showed similar reduced proliferation in the presence of the AhR antagonist CH223191 (Fig 4A, red line). This reduced proliferation was reflected in a decrease in dilution of the cell division tracer CTV dye (Fig 4A), reduced expansion index (Fig 4B), and a reduction in the percentage of divided cells (Fig 4C), while the replication index was similar (Fig 4D), indicating that those $Ahr^{-/-}$ B cells that entered division proceeded through the cell cycle like AhR-sufficient cells. The delay in cell cycle progression was caused by increased retention in G$_0$/G$_1$ phase of AhR-deficient B cells (Fig 4E). We further tested whether AhR deficiency causes apoptosis in B cells cultured for 72 h in medium or IL-4 by staining for annexin-V, but there was no evidence that the reduced expansion of AhR-deficient B cells was linked to apoptosis (Appendix Fig S2).

In order to determine the *in vivo* consequences of impaired proliferation in AhR-deficient B cells, we generated mixed bone marrow chimeras transferring equal numbers of bone marrow cells from $Ahr^{fl/-}$ $mb1^{Cre+}$ mice together with $Ahr^{fl/+}$ $mb1^{Cre+}$ bone marrow (distinguishable by expression of different fluorochromes in the $Rosa26$ locus), into irradiated $Rag1^{-/-}$ hosts. The relative contribution of AhR-deficient B cells (white) vs. AhR-sufficient B cells (black) to the B-cell pool was determined 8 weeks later. As shown in Fig 5A and B, the composition of the mature B-cell pool was substantially skewed in favour of B cells that originated from $Ahr^{fl/+}$ $mb1^{Cre+}$ bone marrow, whereas bone marrow-resident immature subsets of B cells did not show this trend.

To test whether AhR regulates proliferation in response to antigenic stimulation, we co-transferred splenocytes from $SW_{HEL}$ $Ahr^{-/-}$ and $SW_{HEL}$ $Ahr^{+/+}$ mice (distinguishable by CD45 allotype expression) into C57Bl/6 CD45.1$^+$ hosts together with SRBC-HEL (Fig 5C). 7 days later, the distribution of HEL-specific AhR-deficient and AhR-sufficient B cells in the spleen of recipient mice was analysed. Immunization with SRBC-HEL caused expansion of both $Ahr^{+/+}$ (black, CD45.1$^+$ CD45.2$^+$) and $Ahr^{-/-}$ (white, CD45.2$^+$) B cells compared to the mock immunized controls (Fig 5D). However, AhR-sufficient B cells expanded more than AhR-deficient B cells so that over 60%

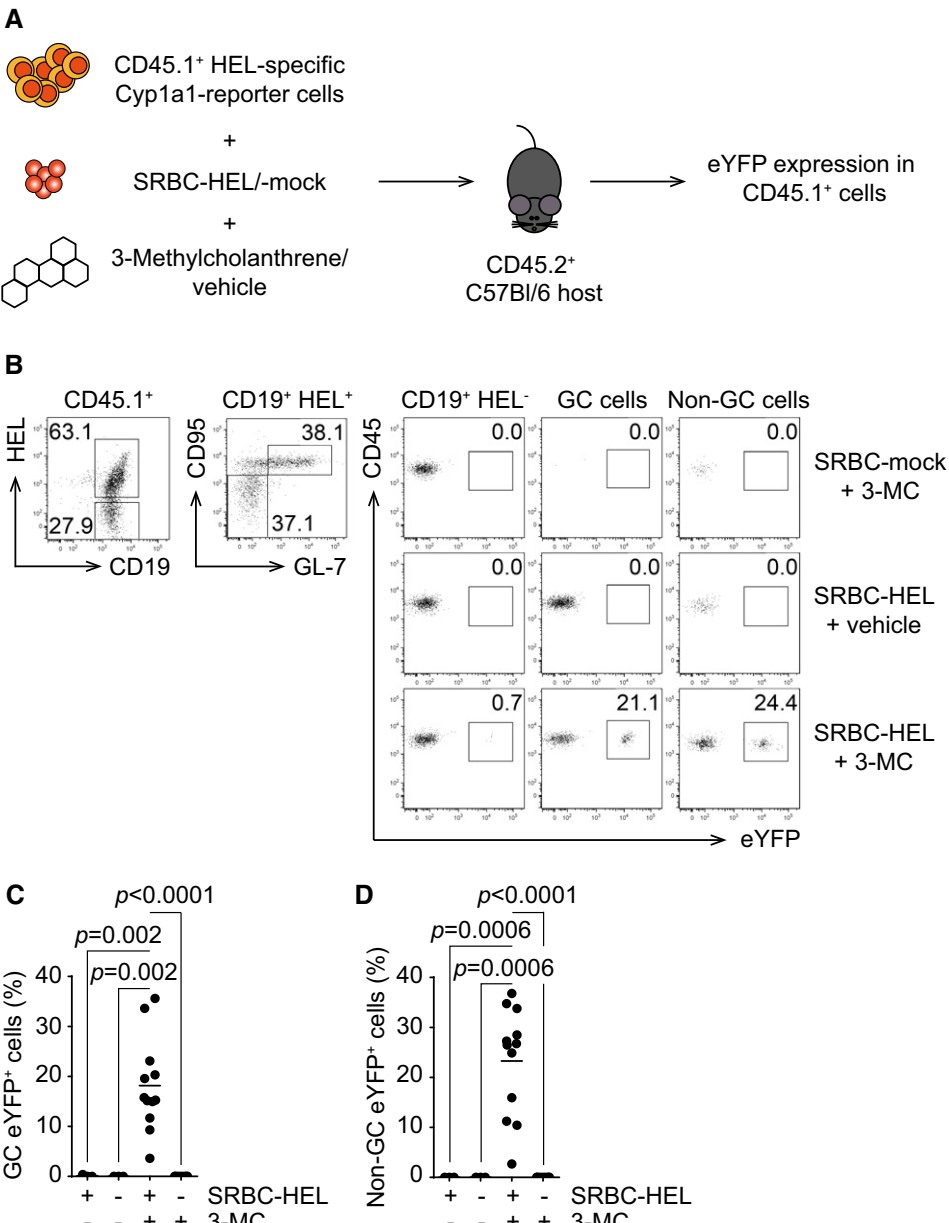

**Figure 3.  B-cell activation in the presence of 3-MC promotes *Cyp1a1* transcription *in vivo*.**

A   Host C57Bl/6 mice were injected with CD45.1$^+$ splenocytes, isolated from $SW_{HEL}$ × $Cyp1a1^{Cre}$ R26R eYFP mice together with SRBC-HEL or SBRC-mock. The transferred mice were treated with 3-MC or vehicle. Readout at d7 post-challenge was eYFP expression in transferred CD45.1$^+$ cells.

B   Representative flow cytometry analysis of Cyp1a1 expression, reported by eYFP, in CD45.1$^+$ cells harvested from mice challenged as in (A). Gating is indicated above the plots. Representative data of $n$ = 3 independent experiments. GC cells: CD95$^+$ GL-7$^+$; non-GC cells: CD95$^-$ GL-7$^-$.

C, D   Summary of percentage of GC (C) and non-GC eYFP$^+$ (D) cells from flow cytometry analysis. $n$ = 3 independent experiments; lines indicate mean; one-way ANOVA, Tukey's multiple comparison test.

of the HEL-specific B cells detected in the recipient mice originated from the AhR-sufficient input, whereas AhR-deficient B cells only represented less than 40% of the HEL-specific B-cell pool (Fig 5E).

This confirmed that AhR-deficient B cells are less prone to proliferate in response to BCR engagement and lose out in competition with AhR-sufficient B cells.

To test whether the reduced proliferation potential of $Ahr^{-/-}$ B cells has further functional consequences *in vivo*, we tested whether

AhR deficiency had an impact on the generation of short-lived splenic and long-lived bone marrow plasma cells (PCs) in non-immune mice. $Ahr^{fl/-}$ $mb1^{Cre+}$ mice showed a reduced number of PCs in the spleen as compared to AhR-sufficient controls (Appendix Fig S3A and B). On the other hand, in the bone marrow long-lived PCs were represented in similar numbers in $Ahr^{fl/-}$ $mb1^{Cre+}$ mice compared with $Ahr^{fl/+}$ $mb1^{Cre+}$ mice (Appendix Fig S3C and D). Immunization with NP-CGG, however, did not reveal

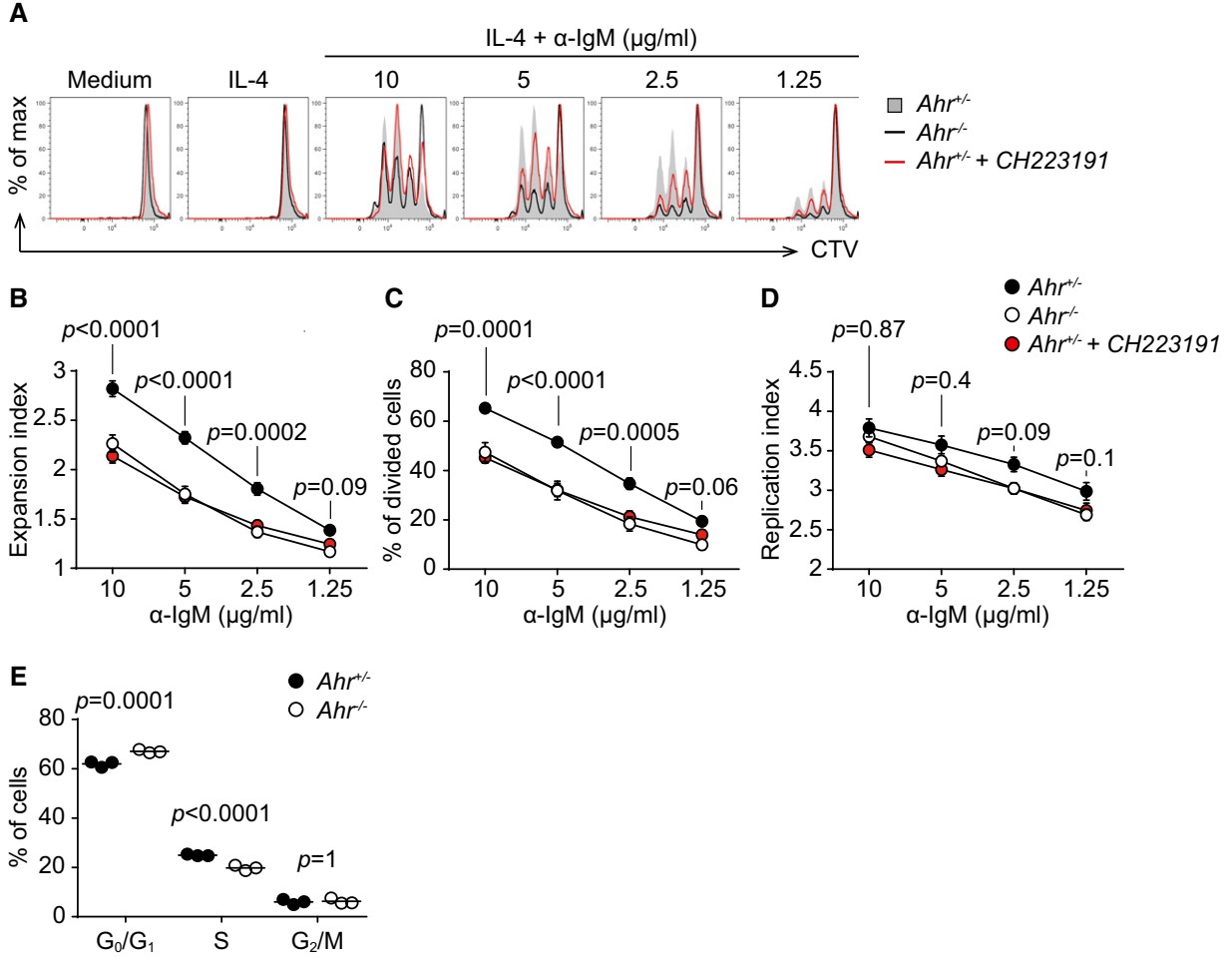

**Figure 4. AhR deficiency impairs B-cell proliferation *in vitro*.**

A    Flow cytometry analysis of CTV dilution in splenic CD19+ cells sorted from non-immune $Ahr^{fl/+} mb1^{Cre+}$ ($Ahr^{+/−}$ B cells, solid grey) and $Ahr^{fl/−} mb1^{Cre+}$ ($Ahr^{−/−}$ B cells, black) mice stimulated for 72 h as indicated. Treatment of $Ahr^{+/−}$ CD19+ cells with CH223191 is indicated in red. Representative data of $n = 4$–5 independent experiments.

B–D    Flow cytometry analysis of expansion index (B), % of divided cells (C) and replication index (D) of splenic CD19+ cells sorted from non-immune $Ahr^{fl/+} mb1^{Cre+}$ (black) and $Ahr^{fl/−} mb1^{Cre+}$ (white) mice stimulated for 72 h as indicated. Treatment of $Ahr^{+/−}$ CD19+ cells with CH223191 is indicated in red. $n = 4$–5 independent experiments; mean ± SEM; two-way ANOVA, Sidak's multiple comparison test.

E    Flow cytometry analysis of cell cycle distribution of splenic CD19+ cells purified from non-immune $Ahr^{fl/+} mb1^{Cre+}$ (black) and $Ahr^{fl/−} mb1^{Cre+}$ (white) mice stimulated for 48 h with 5 μg/ml α-IgM. Data representative of $n = 2$ independent experiments; lines indicate mean; two-way ANOVA, Sidak's multiple comparison test.

any difference between splenic PC numbers of $Ahr^{fl/−} mb1^{Cre+}$ mice compared with $Ahr^{fl/+} mb1^{Cre+}$ mice. This might indicate that the partial proliferative defect can eventually be corrected provided there are not wild-type competitor B cells around (Appendix Fig S3E).

We next determined whether AhR deficiency in B cells influences the generation of high-affinity antibodies *in vivo*. For this, we co-transferred in a 1:1 ratio splenocytes from $SW_{HEL}$ mice, which were either AhR-deficient (white) or AhR-sufficient (black). The hosts were immunized with SRBC coupled with a modified HEL protein carrying three point mutations (HEL3x), for which the $SW_{HEL}$ B-cell receptor has 10,000-fold lower affinity compared with unmodified HEL (Fig EV5A). Therefore, only B cells that have undergone affinity maturation in GC are able to bind HEL3x (Paus *et al*, 2006). AhR deficiency did not compromise the intrinsic ability to undergo affinity maturation as suggested by the similar proportions of

HEL3x-binding IgG1+ or IgG1− B cells between the two genotypes (Figs 5F and EV5B). Nevertheless, when total numbers of B cells that underwent affinity maturation were taken into consideration, both IgG1+ and IgG1− HEL3x-binding B cells from AhR-deficient hosts were substantially reduced compared with their AhR-sufficient counterpart, likely due to their reduced expansion potential (Fig 5G).

Thus, AhR deficiency impairs antigen-dependent B-cell proliferation and the generation of high-affinity antibodies *in vivo*, without compromising the process of affinity maturation itself.

### Cyclin O is defective in AhR-deficient B cells

A role for AhR in the regulation of the cell cycle has been described before, albeit in cell lines, and the suggestion was that induction of

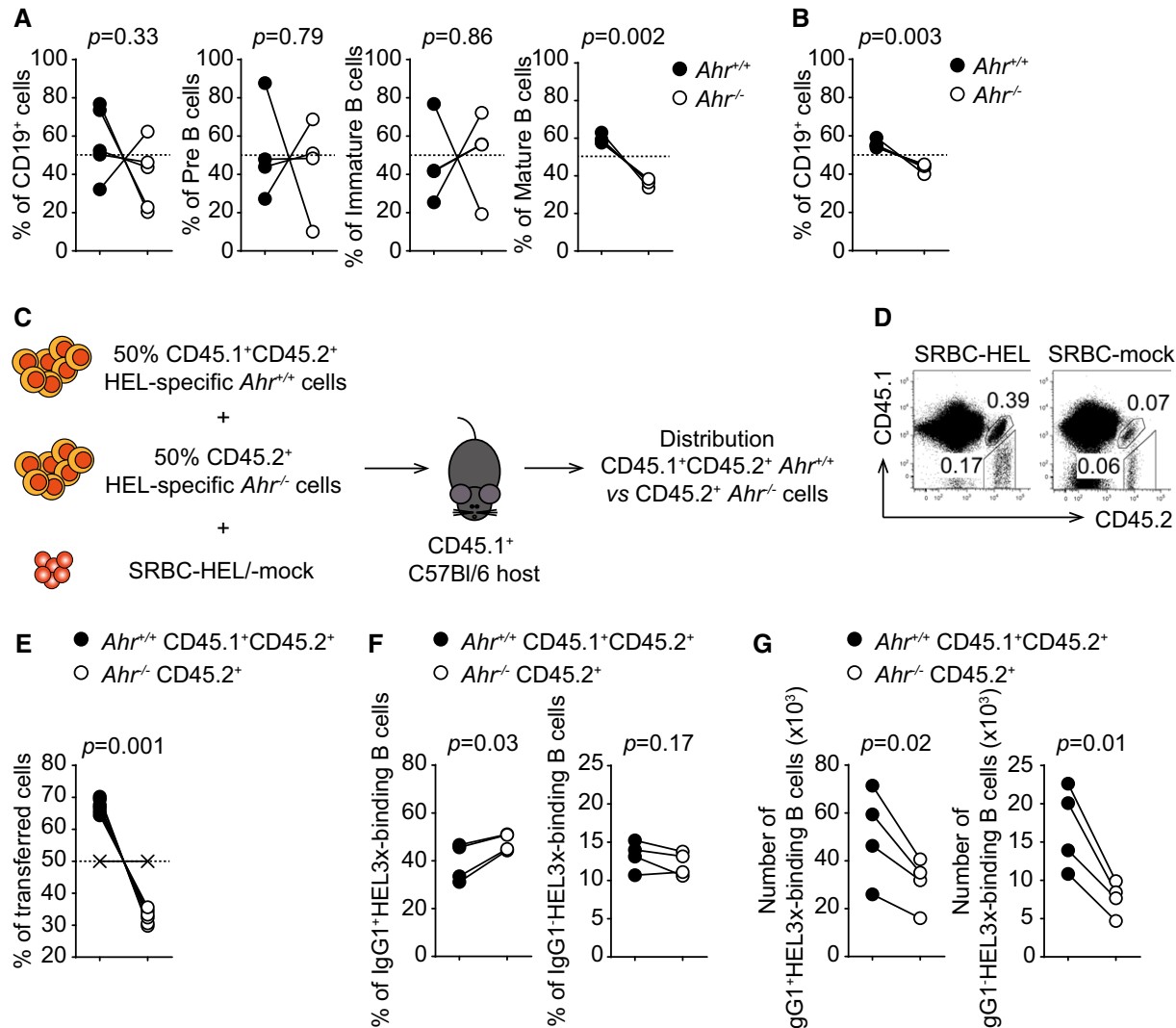

**Figure 5. AhR deficiency impairs BCR-dependent B-cell proliferation *in vivo*.**

A, B  Flow cytometry analysis of distribution of CD45.1⁺ and CD45.2⁺ cells in indicated cell subsets purified from bone marrow (A) and lymph node (B) of sublethally irradiated *Rag1⁻/⁻* mice 8 weeks after reconstitution with equal numbers of bone marrow cells from *Ahr⁺/⁺* (CD45.1⁺) and *Ahr⁻/⁻* (CD45.2⁺) mice. Dashed lines indicate 50% level. Representative data of *n* = 2 independent experiments; two-tailed paired *t*-test.

C  Host *CD45.1* mice were co-transferred with a 1:1 mixture of HEL-specific *Ahr⁺/⁺* CD45.1⁺CD45.2⁺ splenocytes isolated from *SW_HEL Ahr⁺/⁺* mice and HEL-specific *Ahr⁻/⁻* CD45.2⁺ splenocytes isolated from *SW_HEL Ahr⁻/⁻* mice in the presence of SRBC-HEL or SRBC-mock. Readout at d7 post-challenge was distribution of *Ahr⁺/⁺* CD45.1⁺CD45.2⁺ vs. *Ahr⁻/⁻* CD45.2⁺ cells.

D  Flow cytometry analysis of distribution of CD45.1⁺CD45.2⁺ and CD45.2⁺ cells harvested from host mice challenged as indicated above the dot plots. Representative data of *n* = 3 independent experiments.

E  Flow cytometry analysis of distribution of *Ahr⁺/⁺* CD45.1⁺CD45.2⁺ (black) and *Ahr⁻/⁻* CD45.2⁺ (white) cells harvested from host mice. Indicated distribution is quantified relative to transferred cells. X symbols indicate SRBC-mock-treated control. Dashed line indicates 50% threshold. Representative data of *n* = 3 independent experiments; two-tailed paired *t*-test.

F, G  Flow cytometry analysis of distribution (F) and cell numbers (G) of IgG1⁺ and IgG1⁻ HEL3x-binding *Ahr⁺/⁺* CD45.1⁺CD45.2⁺ (black) and *Ahr⁻/⁻* CD45.2⁺ (white) cells harvested from host mice. Representative data of *n* = 2 independent experiments; two-tailed paired *t*-test.

the cell cycle inhibitor p27kip1 underlies the $G_0/G_1$ cell cycle arrest following dioxin treatment, whereas other studies described cell cycle arrest in AhR-deficient cell types (Ma & Whitlock, 1996; Elizondo *et al*, 2000; Levine-Fridman *et al*, 2004). We therefore tested expression of p27kip1, expression of the activation markers CD69, CD86 and MHC II, and calcium mobilization following activation of AhR-sufficient and AhR-deficient B cells with α-IgM. *Ahr⁻/⁻* B cells did not show any alteration in p27Kip1 expression or

activation markers as compared to AhR-sufficient B cells upon BCR engagement (Appendix Fig S4A–D) nor did they show altered calcium mobilization (Appendix Fig S4E). Since AhR targets a large number of genes, we performed an unbiased RNA sequencing screen, comparing *Ahr⁺/⁺* and *Ahr⁻/⁻* B cells activated *in vitro* with α-IgM and IL-4 in the presence of FICZ. Figure 6A and B, and Appendix Tables S1 and S2 show the top 20 differentially regulated genes upon deletion of *Ahr*. Differentially regulated genes were

**A**

| Gene symbol | Avg read count Ahr+/+ | Ahr-/- | Fold change Ahr+/+ vs Ahr-/- | Adjusted p value |
|---|---|---|---|---|
| *Cyp1b1* | 207 | 0 | - infinite | $7.23 \times 10^{-19}$ |
| *Myl10* | 8 | 0 | - infinite | 0.00236 |
| *Tmprss4* | 9 | 0 | - infinite | $2.15 \times 10^{-5}$ |
| *Cyp1a1* | 4733 | 5 | -979 | $3.34 \times 10^{-114}$ |
| *Ahrr* | 297 | 2 | -178 | $1.85 \times 10^{-74}$ |
| *Ccno* | 94 | 1 | -119 | $2.95 \times 10^{-42}$ |
| *Sema3b* | 44 | 1 | -58 | $2.82 \times 10^{-15}$ |
| *Rtn4rl2* | 12 | 0 | -56 | 0.00043 |
| *Fhod3* | 14 | 0 | -52 | 0.00196 |
| *Mpp2* | 32 | 1 | -50 | $3.39 \times 10^{-13}$ |
| *Asb2* | 334 | 8 | -44 | $7.1 \times 10^{-12}$ |
| *Gm15880* | 9 | 0 | -42 | 0.00019 |
| *1700030C10Rik* | 33 | 1 | -39 | $1.71 \times 10^{-12}$ |
| *Ovol1* | 22 | 1 | -31 | $2.23 \times 10^{-5}$ |
| *Muc19* | 14 | 1 | -25 | $3.71 \times 10^{-5}$ |
| *Hic1* | 2293 | 98 | -23 | 0 |
| *Nqo1* | 1102 | 49 | -23 | $6.29 \times 10^{-266}$ |
| *Pltp* | 1277 | 59 | -22 | $7.49 \times 10^{-172}$ |
| *Tiparp* | 21667 | 1098 | -20 | 0 |
| *Bfsp1* | 18 | 1 | -19 | $1.51 \times 10^{-5}$ |

| *Hprt1* | 5510 | 5546 |
|---|---|---|

**B**

| Gene symbol | Avg read count Ahr+/+ | Ahr-/- | Fold change Ahr+/+ vs Ahr-/- | Adjusted p value |
|---|---|---|---|---|
| *Kifc3* | 62 | 257 | 4 | $3.22 \times 10^{-23}$ |
| *Chdh* | 6 | 27 | 4 | $5.46 \times 10^{-5}$ |
| *Fam83f* | 37 | 159 | 4 | $4.08 \times 10^{-15}$ |
| *Gpr35* | 8 | 37 | 4 | $8.83 \times 10^{-4}$ |
| *Mcam* | 7 | 32 | 4 | $7.05 \times 10^{-5}$ |
| *Stard13* | 3 | 16 | 5 | $8.17 \times 10^{-4}$ |
| *Klk1* | 8 | 37 | 5 | $8.62 \times 10^{-7}$ |
| *Siglech* | 51 | 243 | 5 | $5.6 \times 10^{-39}$ |
| *Lair1* | 11 | 55 | 5 | $1.34 \times 10^{-6}$ |
| *Fabp4* | 5 | 26 | 5 | $6.11 \times 10^{-4}$ |
| *3830403N18Rik* | 6 | 32 | 5 | $4.78 \times 10^{-6}$ |
| *Irs1* | 13 | 64 | 5 | $1.47 \times 10^{-15}$ |
| *Npl* | 2 | 13 | 5 | $6.09 \times 10^{-3}$ |
| *Cd209a* | 6 | 33 | 6 | $4.7 \times 10^{-5}$ |
| *Ccl17* | 7 | 47 | 6 | $6.02 \times 10^{-7}$ |
| *Tmem221* | 2 | 11 | 7 | $6.16 \times 10^{-4}$ |
| *Gm15645* | 81 | 1048 | 13 | $5.19 \times 10^{-190}$ |
| *Cd209d* | 1 | 16 | 13 | $4.42 \times 10^{-7}$ |
| *4930447C04Rik* | 2 | 34 | 16 | $5.27 \times 10^{-7}$ |
| *Pxdn* | 0 | 5 | Infinite | $8.21 \times 10^{-3}$ |

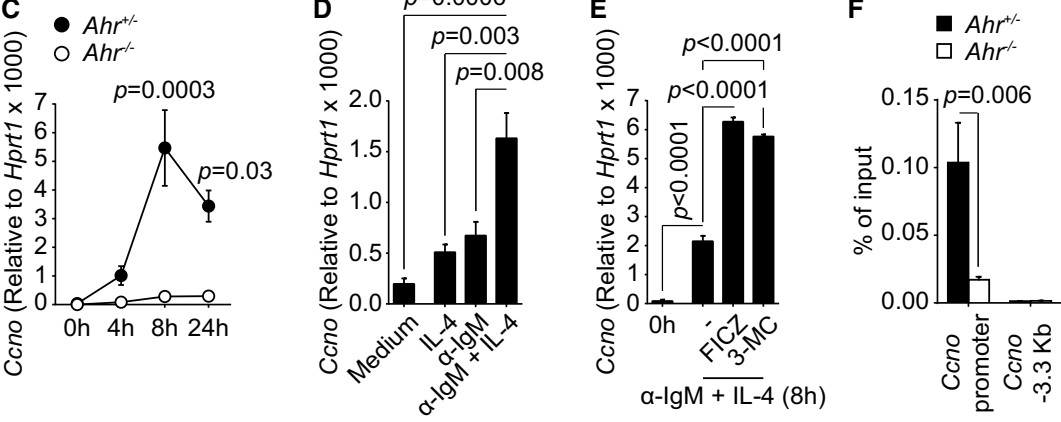

**Figure 6. AhR-deficient B cells fail to up-regulate *Ccno* expression.**

A, B Tables showing the top 20 down-regulated (A) and up-regulated (B) genes in $Ahr^{-/-}$ B cells as compared to $Ahr^{+/+}$ cells, after analysis by RNA sequencing. B cells were activated with 10 μg/ml α-IgM + 20 ng/ml IL-4 for 8 h; 250 nM FICZ was added to the culture during the last 4 h. The average read counts are directly proportional to the extent of expression of a given gene in $Ahr^{+/+}$ or $Ahr^{-/-}$ cells. Data from n = 3 mice per group. Average read counts for the housekeeping gene *Hprt1* in both $Ahr^{+/+}$ and $Ahr^{-/-}$ B cells are indicated below the table in (A).

C qPCR analysis of *Ccno* expression in splenic CD19+ cells isolated from non-immune $Ahr^{fl/+}$ $mb1^{Cre+}$ ($Ahr^{+/-}$ B cells, black) and $Ahr^{fl/-}$ $mb1^{Cre+}$ ($Ahr^{-/-}$ B cells, white) mice, stimulated with 10 μg/ml anti-IgM for the indicated time points. *Ccno* expression was normalized to *Hprt1*. n = 3 independent experiments; mean ± SEM; two-way ANOVA, Sidak's multiple comparison test.

D qPCR analysis of *Ccno* expression in splenic CD19+ cells isolated from C57Bl/6 mice and cultured for 8 h with 20 ng/ml IL-4 and/or 10 μg/ml α-IgM. *Ccno* expression was normalized to *Hprt1*. n = 3 independent experiments; mean ± SEM; one-way ANOVA, Tukey's multiple comparison test.

E qPCR analysis of *Ccno* expression in splenic CD19+ cells isolated from C57Bl/6 mice and cultured for the indicated time points with 20 ng/ml IL-4 and/or 10 μg/ml α-IgM in the presence or absence of 250 nM FICZ or 1 μM 3-MC. *Ccno* expression was normalized to *Hprt1*. n = 3 independent experiments; mean ± SEM; one-way ANOVA, Tukey's multiple comparison test.

F Chromatin immunoprecipitation (ChIP) analysis of AhR interaction with the *Ccno* promoter and an irrelevant region (−3.3 kb from *Ccno* transcription starting site) in $Ahr^{+/+}$ (black) and $Ahr^{-/-}$ (white) B cells 5 h after activation with 10 μg/ml α-IgM + 20 ng/ml IL-4. 250 nM FICZ was added in the last hour of culture. Representative data of n = 2 independent experiments; mean ± SEM; two-way ANOVA, Sidak's multiple comparison test.

screened with gene ontology and pathway analysis tools. Apart from the expected absence of the prototypical AhR target genes *Cyp1a1*, *Cyp1b1* and *Ahrr*, one of the most strongly down-regulated genes in AhR-deficient B cells was *Ccno*, encoding cyclin O, a member of the cyclin family. To confirm the RNA sequencing result and to dissect the AhR dependency of *Ccno* expression dynamics in B cells, splenic CD19$^+$ B cells were isolated from *Ahr*$^{fl/-}$ *mb1*$^{Cre+}$ and *Ahr*$^{fl/+}$ *mb1*$^{Cre+}$ mice, stimulated with α-IgM and *Ccno* expression was assessed over a 24 h period. *Ccno* was maximally induced about 8 h after stimulation in AhR-sufficient B cells, but not detectable in AhR-deficient B cells, confirming the RNA sequencing result (Fig 6C). Furthermore, *Ccno* expression was enhanced by concomitant stimulation with anti-IgM and IL-4 presumably due to the increased levels of AhR under these conditions (Fig 6D). This was further enhanced in the presence of AhR ligands FICZ and 3-MC (Fig 6E). ChIP PCR established that AhR directly bound to the *Ccno* promoter to regulate its expression (Fig 6D).

## Discussion

With the appreciation of physiological roles of the AhR beyond the detoxification of environmental pollutants, immunologists are increasingly focusing on the functions of this transcription factor in different immune cell populations. We had initially described expression of AhR in T cells, where it is confined to the T$_H$17 cell subset and required for the induction of interleukin 22 (Veldhoen *et al*, 2008), but it appears that AhR is more widely expressed in other immune cell types (Stockinger *et al*, 2014). B cells show universal expression albeit at generally low levels, with the exception of subsets like MZB cells and B1 B cells. These subsets are considered to be rapidly responsive to repetitive epitopes displayed by environmental pathogens and self-antigens and may therefore be in an elevated state of activation (Martin & Kearney, 2002; Baumgarth, 2011). Since BCR activation results in up-regulation of *Ahr*, it is likely that the increased *Ahr* expression in MZB and B1 cells mirrors their activation profile rather than representing a characteristic feature. Furthermore, the ability to up-regulate *Ahr* expression upon BCR activation applied to all splenic B-cell subsets, confirming that *Ahr* up-regulation may reflect BCR engagement and is not an exclusive feature of specific B-cell subsets. The link between BCR engagement and *Ahr* induction is still unknown, but does not involve the canonical NF-κB pathway in contrast to what was recently postulated in fibroblasts (Vogel *et al*, 2014).

The role for AhR in B cells has been studied extensively in the context of environmental chemical-mediated toxicity and suppression of the humoral immune response by dioxin is a well-established phenomenon, and exposure of primary B cells or B-cell lines to chemical AhR ligands indicated a role in the control of B-cell response acting at multiple levels (Sulentic & Kaminski, 2011; Sherr & Monti, 2013). Our study of B-cell function in mice in which AhR deficiency was targeted to B cells did not indicate any alterations in B-cell responses against T-independent and T-dependent challenges.

However, upon closer scrutiny it was apparent that AhR-deficient B cells had reduced proliferation potential due to their retention in G$_0$/G$_1$ stage of the cell cycle. This defect was partial, as those B cells that managed to overcome the block proliferated normally. Nevertheless, it was apparent that AhR-deficient B cells lost out in

competition with AhR-sufficient B cells, both in settings of homeostatic proliferation (Fig 5A and B) and upon antigen-driven proliferation (Fig 5D and E). Interestingly, *Ahr* deletion did not impair the ability of B cells to proliferate *per se*, but instead compromised their likelihood to commence the cell cycle. Indeed, the reduced expansion potential of *Ahr*$^{-/-}$ B cells was mirrored by the reduced fraction of cells undergoing cell division. On the other hand, the replication index, indicating the proliferative capacity of cells that already underwent division, was comparable between AhR-sufficient and AhR-deficient B cells. These results suggest that AhR could be a modulator of the B-cell activation threshold, augmenting the sensitivity of B cells to BCR triggers.

B-cell proliferation is important to sustain the affinity maturation process in GC that allows generation of high-affinity antibodies. Recent studies showed that the magnitude of T cell help received by GC B cells in the light zone is positively correlated with the division rate and number of point mutations accumulated in immunoglobulin genes in the dark zone (Victora *et al*, 2010; Gitlin *et al*, 2014). In the *SW*$_{HEL}$ model, AhR deficiency did not impair the intrinsic ability of B cells to undergo affinity maturation, since comparable fractions of *Ahr*$^{+/+}$ and *Ahr*$^{-/-}$ B cells underwent affinity maturation and bound HEL3x. However, because of the expansion defect of *Ahr*$^{-/-}$ B cells as compared to *Ahr*$^{+/+}$ controls, the total number of HEL3x-binding affinity-matured AhR-deficient B cells was lower. Thus, reduced expansion rather than an intrinsic affinity maturation defect led to a reduced ability to generate high-affinity antibodies. These data are in line with the finding that AhR deficiency impacts BCR-driven proliferation, whereas T cell-driven B-cell proliferation that occurs in GC during the affinity maturation process remains unaffected. Indeed, we observed that stimuli other than through the BCR, including CD40 ligation, did not have any effect on *Ahr* expression levels. Mice with AhR-deficient B cells also showed a defect in short-lived plasma cells in the spleen, whereas long-lived plasma cells in the bone marrow appeared normal, suggesting that over time the reduced expansion due to problems of entering into the cell cycle can be compensated.

Thus, it appears that endogenous AhR ligands play a role in maintaining the functional response of B cells to antigen activation. In order to detect AhR activation *in vivo*, we employed a fate reporter mouse model in which mice with Cre recombinase inserted under control of the endogenous *Cyp1a1* promoter were crossed with *Rosa26 eYFP* reporter mice such that cell that has activated the AhR pathway and induced *Cyp1a1* would be permanently labelled with the eYFP fluorochrome (Henderson *et al*, 2015). Under steady-state conditions, B cells of these mice did not express the reporter and in general we found very low eYFP expression, mainly in non-haematopoietic cells (M. Villa, unpublished observation). This is most likely due to the tight regulation of AhR activation that is subject to strong feedback control and therefore might not result in sufficiently prolonged *Cyp1a1* expression to turn on Cre recombinase. However, deliberate addition of AhR agonists either *in vitro* or together with immunization *in vivo* strongly induced this reporter, confirming its functionality. It therefore seems that the AhR pathway is functional in B cells *in vivo*, but exposure to endogenous ligands was subthreshold of detection with the reporter.

Observations in the toxicology field have implicated AhR in cell cycle regulation with different effects in different cell types (Puga *et al*, 2002). Whereas transient AhR engagement in a hepatoma cell

line promoted $G_0/G_1$ to S phase transition, sustained activation with dioxin increased p27Kip1 expression, promoting cell cycle arrest (Levine-Fridman *et al*, 2004). In contrast, other studies showed that AhR deficiency negatively affected cell proliferation (Ma & Whitlock, 1996; Elizondo *et al*, 2000; Tohkin *et al*, 2000). It is possible that prolonged AhR activation via dioxin that is not metabolized might in some cases mimic AhR deficiency, presumably due to the fact that this exposure can lead to profound down-regulation and degradation of AhR (Pollenz *et al*, 1998). It is noteworthy that AhR knockdown was recently shown to reduce expression of E2F1, which is involved in regulation of cell cycle and apoptosis (Frauenstein *et al*, 2013). These observations are difficult to extrapolate to primary B cells and we did not observe any alteration in p27kip1 levels in AhR-deficient B cells.

In order to understand the mechanistic basis for the defective proliferation of AhR-deficient B cells, we carried out an unbiased RNA sequencing screen comparing BCR-activated B cells from AhR-deficient and AhR-sufficient mice. It should be noted that our screen did not overlap with previous screens of dioxin-activated B-cell lines, clearly indicating that the chemical activation of AhR interferes with B-cell responses on a different level (De Abrew *et al*, 2010).

A top scoring differentially expressed gene, apart from AhR pathway genes such as *Cyp1a1* and *Ahrr,* was *Ccno*, encoding the cyclin family member cyclin O, which seemed a plausible candidate to explain the reduced proliferative potential of AhR-deficient B cells and furthermore appears to be directly regulated by AhR as AhR bound to the *Ccno* promoter.

Cyclin O was erroneously identified as a component of the molecular complex involved in antibody class switch recombination and affinity maturation (Muller & Caradonna, 1991). However, a later report described a possible role for cyclin O in controlling cell cycle, as it is expressed as a function of the cell cycle. Cyclin O expression peaked during the $G_1$ phase, while undergoing complete turnover by the end of the cell cycle, an expression pattern typical of cell cycle regulators (Muller & Caradonna, 1993). AhR-deficient B cells failed to express *Ccno,* while it was transiently up-regulated in $Ahr^{+/+}$ B cells about 8 h after activation. Conservation analysis between the human and mouse *Ccno* gene established the presence of multiple dioxin response elements (DREs) within several regions of the *Ccno* locus (Appendix Fig S5). Although the involvement of *Ccno* in regulation of B-cell proliferation is so far not experimentally proven, this molecule represents a potential candidate.

Thus, AhR sensing of endogenous ligands by B cells has a subtle, but detectable influence on their physiology. As the exposure of B cells to endogenous AhR ligands was not measurable using a *Cyp1a1* fate reporter, we can only infer a role for AhR signalling in B cells through the deleterious effects of AhR deficiency. It is likely that the transient nature of AhR signalling is of particular significance for its physiological functions.

# Materials and Methods

## Mice and immunizations

C57BL/6J (in the text called C57Bl/6), *B6.129S7-Rag1^{tm1Mom}/J* (*Rag1^{−/−}*), *B6.129-Ahr^{tm1Bra}/J* (*Ahr^{−/−}*), *Cd79 a^{tm1(cre)Reth}* (*mb1^{Cre}*) (Hobeika *et al*, 2006), *Ahr^{tm3.1Bra}/J R26R eYFP* (*Ahr^{fl/fl}*), *Cyp1a1^{Cre}*

*R26R eYFP* (Henderson *et al*, 2015) and *SW_{HEL}* mice (Phan *et al*, 2003) were bred and kept at Francis Crick Institute animal facilities under specific pathogen-free conditions. Mice were used after 8 weeks of age, age- and gender-matched unless otherwise stated. All animal experiments were performed according to institutional guidelines (Francis Crick Institute Ethical Review Panel) and UK Home Office regulations.

TNP-Ficoll (Biosearch Technologies; 10 μg/mouse in PBS) and NP-CGG (Biosearch Technologies; 10 μg/mouse in PBS:Alum (Thermo Scientific) 3:1) immunizations were done by intra-peritoneal (i.p.) injection. For cholera toxin (Ctx; List Biological Laboratories, Inc.) immunization, mice were starved for 2 h and HBSS + 7.5% sodium bicarbonate solution (4:1 ratio) was administered by intra-gastric (i.g.) gavage, followed 30 min later by i.g. gavage of 2.5 μg active Ctx in PBS. Mice were culled at d14 post-immunization for analysis.

*SW_{HEL}* mice were used as donors of HEL-specific B cells that were transferred into a naive congenic recipient mouse together with HEL conjugated to the carrier SRBC to induce a T-dependent response. Host and donor differed in the allotypic marker CD45, which allowed assessment of donor-specific responses in an immune-competent mouse (Phan *et al*, 2003).

Briefly, 50,000 HEL$^+$ B cells were transferred together with $10^9$ HEL-conjugated SRBC (SRBC-HEL) intravenously (iv) in PBS into recipient mice. For each experiment, a mock control immunized with uncoupled SRBC was used. Mice were analysed at d7 post-immunization. In some experiments, the AhR ligand 3-MC (Sigma) was dissolved in corn oil and injected ip at a concentration of 26.5 mg/kg. For conjugation, SRBCs (Patricell) were extensively washed in PBS and resuspended in conjugation buffer (0.35 M mannitol, 0.01 M NaCl). HEL (2 mg/ml; Sigma) or HEL3x (100 μg/ml; kindly provided by R. Brink) was added 1:10 and incubated 10 min at 4°C. Without washing, EDCI (Novabiochem) 100 mg/ml in conjugation buffer was added 1:10 and incubated 30 min at 4°C. SRBCs were then extensively washed in PBS, counted and resuspended in PBS.

To generate bone marrow chimeras, lethally irradiated (2 × 5 Gy) *Rag1^{−/−}* recipient mice (6–8 weeks old) received iv transfer of bone marrow (BM) cells from gender-matched donors. Recipient mice were kept under antibiotic treatment (Baytril 0.02% in drinking water; Bayer) for 2 weeks. Blood samples were taken 4 weeks after reconstitution to check for chimerism, and recipient mice were used for experiments not earlier than 8 weeks post-reconstitution.

## Flow cytometry

B-cell subsets were sorted or analysed as follows: from spleen: FoB (B220/CD19$^+$ CD93$^-$ IgM$^+$ CD23$^+$); MZB (B220/CD19$^-$ IgM$^{hi}$ CD23$^-$); TrB (B220/CD19$^+$ CD93$^+$); PC (B220$^-$ CD138$^+$). TrB subsets were further separated on the basis of IgM and CD23 expression: T1 (IgM$^+$ CD23$^-$), T2 (IgM$^+$ CD23$^+$) and T3 (IgM$^{lo}$ CD23$^+$). From bone marrow: ProB (B220$^+$ CD2$^-$); PreB (B220$^+$ CD2$^+$ IgM$^-$ IgD$^-$); Immature B (B220$^+$ CD2$^+$ IgM$^+$ IgD$^-$); Mature B (B220$^+$ CD2$^+$ IgM$^+$ IgD$^+$); PC (IgD$^-$ B220$^-$ CD138$^+$); Stem cells (B220$^-$ cKit$^+$); Early ProB (cKit$^{dim}$ B220$^+$); B-cell precursors (cKit$^-$ B220$^+$ CD19$^+$); NK cell precursors (cKit$^-$ B220$^+$ CD19$^-$). From peritoneal cavity: CD5$^+$ B1 (IgM$^{hi}$ CD23$^-$ CD5$^+$); CD5$^-$ B1 (IgM$^{hi}$,

CD23$^-$ CD5$^-$) CD5$^-$ B1; B2 (IgM$^+$ CD23$^+$). From Peyer's patches: GC B (B220/CD19$^+$ CD95$^+$ GL-7$^+$); non-GC B (B220/CD19$^+$ CD95$^-$ GL-7); T$_{FH}$ (CD4$^+$ TCRβ$^+$ PD-1$^+$ CXCR5$^+$).

The following antibodies were used for flow cytometry: CD2, CD4, CD5, CD19, CD23, CD45.1, CD45.2, CD69, CD86, B220, MHC II, PD-1, TCRβ (Biolegend); CD93 (clone AA4.1), cKit, GL-7, IgD (eBioscience); CD95, CXCR5, IgG1, CD138 (clone 281-2) (BD Biosciences); IgM (Jackson Immunores. Lab.; F$_{ab}$ fragment).

Cell proliferation was quantified by flow cytometry using Cell-Trace™ violet cell proliferation kit (Life Technologies). Cell proliferation parameters such as % of divided cells, expansion index and proliferation index were obtained using the proliferation platform of FlowJo software (version 9; Treestar). % of divided cells represents the fraction of the initial population that underwent cell division. Expansion index indicates the fold expansion of the overall culture. Replication index indicates the fold expansion of only the responding cells.

Cell cycle stage was analysed by flow cytometry using Vybrant® DyeCycle™ violet stain (Life Technologies).

For calcium flux analysis, cells were incubated with Indo-1 AM (Invitrogen) IMDM 5% FCS (final Indo-1 AM concentration 2 μM) for 30 min at RT. Cells were washed and resuspended at a concentration of 10$^7$ cells/ml in IMDM 5% FCS and kept at 37°C until analysis. Cells were stimulated with anti-IgM at appropriate concentrations. Calcium flux was quantified by measuring the ratio between the violet and blue emission upon Indo-1 AM excitation by UV laser.

Flow cytometric analysis was performed on BD FACS Canto II, BD LSR II or BD LSRFortessa flow cytometers (BD Biosciences). Cell sorting was performed on a BD FACSAria II, BD Influx (BD Biosciences) or MoFlo XDP (Beckman Coulter). Data were analysed using FlowJo software (Treestar).

## Cell culture

Splenic B cells were purified using the EasySep mouse B-cell isolation kit according to manufacturer instructions. Purity was assessed by flow cytometry, and 90–95% of the purified cells were CD19$^+$. Cells were cultured at appropriate concentrations (i.e., 250,000 cells/well in 200 μl in 96-well plates) in complete IMDM at 37°C, 7% CO$_2$. Cells were treated with α-IgM, concentrations between 10 and 1.25 μg/ml (F$_{ab}$ fragment; Jackson Immunores. Lab.), IL-4 20 ng/ml (R&D Systems), CD40L 20 ng/ml (R&D Systems), LPS 1 μg/ml (Alexis Biochemicals), BAFF 100 ng/ml (Peprotech), CpG 1 μg/ml (ODN1668TypB; Source Bioscience), FICZ 250 nM (Enzo Life Sciences), CH223191 3 μM (Calbiochem), BI605906 10 μM (kindly provided by S. Ley).

## Chromatin immunoprecipitation, RNA extraction, cDNA generation and real-time RT–PCR

Chromatin immunoprecipitation was performed on 10$^7$ cells that were cross-linked with 1% paraformaldehyde. Chromatin was isolated upon sequential incubation steps with the following buffers: buffer 1 (Tris–HCl pH 8 10 mM, EDTA 1 mM, NP40 0.5%, PMSF 1 mM), buffer 2 (Tris–HCl pH 8, EDTA 1 mM, NaCl 0.5 M, Triton X-100 1%, deoxycholate 0.5%, sarcosyl 0.5%, PMSF 1 mM) and buffer 3 (Tris–HCl pH 8 10 mM, EDTA 1 mM, NaCl 100 mM,

sarcosyl 0.1%, PMSF 1 mM). Chromatin was then sheared by sonication. 10% of sonicated cell extract was kept as input. Chromatin was added with RIPA buffer and immunoprecipitated overnight at 4°C with 2 μg of anti-AhR antibody (Enzo Life Sciences, BML-SA210). Protein G Dynabeads (Life Technologies) were then added to the cell extract for 3 h at 4°C. Samples were washed 7× in RIPA wash buffer (added with SDS 0.1%). Crosslinking was reversed in TE buffer pH 8, SDS 0.5%, proteinase K 200 μg/ml and incubated overnight at 65°C. DNA was isolated with phenol/chloroform and analysed by quantitative PCR and by normalization relative to input DNA amount. The following primers were used for the *Ccno* promoter: forward 5′-GGGGCTCAGCCAGTGAGA-3′; reverse 5′-GGCGCAGCTCTAAGTACCC-3′.

RNA was extracted using TRIzol® Reagent (Life Technologies) according to manufacturer instructions. RNA to cDNA conversion was performed using the Omniscript RT kit (Qiagen). Quantification of target genes was done by quantitative PCR using Taqman technology (Applied Biosystems). Reaction mixes were run on the 7900HT Applied Biosystems thermal cycler. TaqMan primer pairs used to quantify target genes were as follows: *Ahr* Mm00478930_m1; *Ccno* Mm01297259_m1; *Cyp1a1* Mm00487217_m1; *Hprt1* Mm00446968_m1 (Applied Biosystems).

## RNA sequencing analysis

RNA was extracted using the RNAeasy mini kit (Qiagen) and following manufacturer instructions. 1 μg RNA was used for further analysis. Library generation was performed according to manufacturer instructions using the TruSeq stranded mRNA library prep kit (Illumina). Libraries were barcoded and run on an Illumina HiSeq 2000.

The RNAseq data are available in the Gene Expression Omnibus (GEO) database (http://www.ncbi.nlm.nih.gov/geo/query/acc.cgi?acc=GSE86521) with accession number GSE86521.

## Nuclear/cytoplasmic protein fractionation and Western blot

10–20 × 10$^6$ cells were incubated in cytoplasmic lysis buffer (HEPES pH 7.6 10 mM, EGTA 0.1 mM, KCl 10 mM, MgCl$_2$ 1.5 mM, dithiothreitol 1 mM, NaF 20 mM, protease inhibitor cocktail (Roche), PMSF 1 mM) for 10 min at 4°C. Detergent NP-40 was added at a final concentration of 0.2%, and cells were incubated for 1 min on ice. Supernatant was collected as cytoplasmic protein fraction. The nuclei pellet was added with nuclear lysis buffer (Tris–HCl pH 7.5 50 mM, NaCl 150 mM, EDTA 2 mM, Triton X-100 1%, dithiothreitol 5 mM, deoxycholate 0.5%, SDS 0.1%, protease inhibitor cocktail, PMSF 1 mM) and incubated 15 min at 4°C. Supernatant was collected as nuclear protein fraction.

For Western blot, following antibodies were used: anti-AhR 1:1,000 (Enzo Life Sciences, BML-SA210); anti-Sam68 1:2,000 (Santa Cruz); anti-tubulin 1:2,500 (Sigma); anti-IκBα 1:1,000 (Santa Cruz); anti-β-actin 1:1,000 (Sigma); anti-Gapdh 1:10,000 (Sigma); anti-p27kip1 1:200 (R&D Systems); anti-mouse 1:10,000 (GE Healthcare); anti-rabbit 1:10,000 (GE Healthcare).

## Enzyme-linked immunoadsorbent assay

Enzyme-linked immunoadsorbent assay (ELISA) to detect anti-TNP, anti-NP, and anti-Ctx antibodies (Southern Biotech) was performed

using TNP-BSA 33 μg/ml (Biosearch Technologies), NP-BSA 10 μg/ml (Biosearch Technologies) and Ctx 10 μg/ml (List Biological Lab.) as coating.

### Statistical analysis

Statistical analysis was performed using GraphPad Prism software. In case of normally distributed samples, two-group comparison was done using a two-tailed unpaired *t*-test, unless in the presence of paired samples (paired *t*-test). Comparisons among more than two groups were analysed with one- or two-way ANOVA, followed by multiple comparison correction (Dunnett, Sidak or Tukey tests). Statistical significance is indicated as precise *P*-value.

Expanded View for this article is available online.

### Acknowledgements

This work was supported by the Francis Crick Institute, which receives its core funding from Cancer Research UK, The UK Medical Research Council and the Wellcome Trust. We would like to acknowledge the animal facility for expert management of breeding and maintenance of our mice as well as support by the Flow Cytometry facility and the Sequencing Facility. This work was funded by a Boehringer Ingelheim Fonds PhD Fellowship to M.V., a Wellcome Investigator Grant to B.S. and Cancer Research Programme Grant C4639/A10822 to C.J.H. and C.R.W.

### Author contributions

MV performed experiments, analysed data and wrote the manuscript. BS conceived the study, wrote the manuscript and secured funding. MG and MT performed specific experiments, and HA performed the bioinformatic analysis of the RNAseq data. CJH, CRW and RB provided reagents and expert advice.

### Conflict of interest

The authors declare that they have no conflict of interest.

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
