## [Review Process File · The EMBO Journal]

Manuscript EMBO-2016-95027

Aryl hydrocarbon receptor is required for optimal B-cell proliferation

Matteo Villa, Manolis Gialitakis, Mauro Tolaini, Helena Ahlfors, Colin J Henderson, C Roland Wolf, Robert Brink, Brigitta Stockinger

Corresponding author: Brigitta Stockinger, The Francis Crick Institute

Review timeline:

Submission date:	15 June 2016
Additional Correspondence:	28 July 2016
Editorial Decision:	15 August 2016
Revision received:	05 September 2016
Editorial Decision:	04 October 2016
Revision received:	05 October 2016
Accepted:	11 October 2016

Editor: Karin Dumstrei

Transaction Report:

Additional Correspondence

28 July 2016

Thank you for submitting your manuscript to The EMBO Journal. Your study has now been seen by three referees and their comments are provided below.

As you can see below your manuscript received a bit of a mixed response. The referees appreciate the analysis, but also find that it needs to be significantly extended for consideration here.

Given the comments raised and as it is not clear if the manuscript can be sufficiently extended, I would like to ask you to provide me with a point-by-point response with what can be done within a timeframe of 3-6 months. Based on this I will take the decision on the manuscript. Please also take into consideration the comments raised in the referees' general comments.

I am going away for a short vacation from tomorrow and will be back on the 8th of August. I will take a look at your response as soon as I am back.

Sorry for the delay in getting the decision back to you, but I have just received the third referee report today.

REFeree REPORTS

Referee #1

The manuscript of Villa et al. studies mice with the B cell specific deletion of the gene that codes for the ligand-activated transcription factor aryl hydrocarbon receptor (AhR). The AhR gene is broadly albeit rather weakly expressed in B cells including CD5 positive B1 cells and plasma cells as has been shown previously (Sherr and Monti 2013). The authors then confirm previous studies that IL4 can increase the expression of AhR. As a ligand-dependent nuclear transcription factor, AhR is activated by environmental chemicals such as dioxin derivatives. Using a YFP reporter located inside the AhR-regulated gene Cyp1a1 the authors show that stimulation of B cells with IL4 and anti-IgM can increase YFP expression in ex vivo cultured B cells. The expression of this reporter gene, however, is rather low in mice treated with the AhR ligand 3-methylcholanthrene (3-MC) alone. It is only seen when these mice are immunized at the same time. In Fig. 4 the authors finally study the B-cell specific AhR-deficient mice and show that B cells in these mice expand less well than wildtype B cells when cultured with IL-4 plus anti-IgM. Similar data were also obtained in a competitive transfer experiment and they show that the AhR-deficient mice have fewer short-lived plasma cells. In a comparative transcriptome analysis of AhR-deficient and AhR-sufficient B cells the authors identified a small down-regulation of the cell cycle regulator cyclin O (Ccn0) and they suggest that the reduced expression could be responsible for this reduced expansion of AhR-deficient B cells in culture and in vivo. However, as they mention at the end of their manuscript, they were unable to rescue this phenotype by Ccn0 overexpression.

Major points:

The major problem of this manuscript is that one learns little new about the role of AhR in normal B cell development and function and that cellular signaling pathways where AhR plays a role remain ill-defined. Specifically, as AhR is a ligand-dependent receptor, one has to assume that, without the exposure to toxins, there must be a natural ligand, but the nature of this ligand remains obscure. Here, it would be important to clarify whether AhR indeed needs ligand binding or whether it can also work without a ligand. Based on data shown in Fig. 2B, the second option is rather unlikely. Furthermore, the results presented in Fig.3B suggest an absence of a natural ligand in vivo, considering that the expression of the Cyp1a1 reporter is not induced unless the mice are treated with an external AhR ligand. Identifying a natural ligand or providing proof that AhR is activated in B cells in vivo without the introduction of an external AhR ligand is crucial for this study given that the aim of this manuscript is to analyze "the impact of AhR deficiency on B cell function in the absence of xenobiotic influences" Another problem of this manuscript is that the claim made in the title of the manuscript (optimal B cell proliferation) is not fully explored. The reduced expansion of AhR-deficient B cells observed in vivo and in vitro could be also due to increased apoptosis. That is not excluded currently in the manuscript. Furthermore, and more importantly, the mechanism how AhR regulates proliferation remains ill defined.

Specific comment:

In Fig. 1B the authors confirm previously published data that exposure of B cells to IL-4 increases AhR expression and the same is true with anti-IgM treatment. As in the later experiments they often use a combination between anti-IgM and IL-4, it would be important to show here whether this treatment also has a synergetic effect on the expression of the AhR gene. Furthermore, it would be more appropriate not only to show transcription, but also protein expression data in this study. This manuscript completely lacks a phenotypic analysis of the B cell compartment in the B cell specific AhR knockout mice. Is the development of these B cells in the bone marrow and in the periphery completely normal? If not, and if there is an altered B cell compartment for example in the spleen, then this could also explain the differences in the expansion of these cells in ex vivo cultures or in vivo transfer experiments. Thus, these data should at least be mentioned if not shown in the supplement by the authors. It would also be important to know whether the phenotype of this mouse is in any way different to the mice with a complete AhR knockout which have been published previously.

In Fig.2C it would be helpful to include AhR deficient B cells to prove that the expression of the Cyp1a1 reporter is truly AhR dependent.

In Fig.5 the authors find AhR deficient B cells to be outcompeted by wild type B cells when transferred into mice, which is attributed to impaired proliferation of AhR deficient B cells. However in 5A the phenotype is only observed in the population of mature B cells. Homeostatic proliferation usually does not significantly contribute to the maintenance of the mature B cell pool. Thus additional experiments addressing the proliferation, survival and maturation of AhR deficient B cells in vivo would be needed to help explain the observed phenotype.

Fig.6 Why were non immunized mice used for this experiment? Is the decrease in splenic PCs also

observed after a thymus dependent or thymus independent immunization? What is the marginal zone phenotype of these mice? Are the reduced PC numbers a result of decreased marginal zone B cell numbers or, as stated in the manuscript, a result of AhR having an "impact on B cell proliferation"? In Fig. 7 the authors study the expression of the AhR target gene *Ccno* in AhR-sufficient and AhR-deficient B cells stimulated with anti-IgM. Is this expression increased by anti-IgM + IL-4 treatment and in the presence of the toxin ligand?

Referee #2:

In this manuscript the authors explore the function of the aryl hydrocarbon receptor (AhR) in B cells. The receptor, a ligand-dependent transcription factor, was initially characterized as a sensor for chemical pollutants and to date the physiological ligand *in vivo* is not known. The authors present convincing data using a combination of AhR-deficient mice, AhR-reporter mice and chimeric mice that an AhR-deficiency results in reduced antigen-driven B cell proliferation, although the results are not dramatic, and suggest cyclin O, a target of AhR, contributes to the deficiency. These are novel findings that contribute to an understanding of the impact of an inability to sense AhR ligands on B cell responses to antigen.

What I found most interesting about the study was the effect of BCR ligation and IL-4 treatment on the response to AhR agonist. What we learn is that BCR crosslinking and IL-4 treatment increase the transcription of *Ahr*. However, the authors don't provide any insight as to the repercussion of increased levels of *Ahr* transcripts. For example, do BCR crosslinking or IL-4 treatment result in increased sensitivity to AhR agonist? BCR crosslinking does not appear to induce AhR agonists as in the absence of an exogenous agonist AhR is not translocated to the nucleus and does not induce *Cyp1a1* expression. However, there appears to be a peculiar phenomenon that suggest that B cells may be communicating with each other, namely that the fold increases in *Ahr* (Fig. 1A,B) and *Cyp1a1* (Fig. 2B,D) are significantly greater when total CD19⁺ cells are analyzed versus individual B cell subpopulations. As the authors point out the B cells appear to respond to tryptophan products in the culture media. Is it possible that B cells produce metabolic products upon BCR crosslinking that alter the response of neighboring cells?

Referee #3

In this paper by Villa et al., the authors study the role of the aryl hydrocarbon receptor (AHR) in murine B cell maturation, proliferation, and affinity maturation.

They first demonstrate constitutive and inducible expression of AHR and its activation by agonists, using *cyp1a1* induction as a read-out. They then perform a series of *in vivo* studies, using several highly sophisticated mouse models of conditional B-cell specific AhR-deficiency in combination with a *cyp*-reporting system. These experiments show that AhR presence or absence results changes in B cell proliferation upon antigenic challenge. Following up on this, they try to identify the responsible factors by gene expression profiling of AhR-deficient B cells. They report a strong down-regulation of cyclin O, and link this to the observed low proliferation of B cells.

Overall this is a complex and well-performed study, which addresses an up-to-date topic. While research on the role of AhR in the immune system has focused in particular on T cells and innate immune cells, there are only few studies on B cells. Moreover, many of these studies deal with B cell lines, not primary B cells.

Nonetheless, the manuscript suffers somewhat from over-interpretation, especially regarding the mechanistic part of identifying the down-stream events regarding proliferation. This part appears still a bit immature, and indeed the authors say that they could not repeat one experiment up to now. This experiment should thus not be used to incept conclusions.

While the final experiment with unbiased RNA sequencing to identify relevant genes is useful, it is surprising that the authors do not connect their findings/lack of their finding to knowledge regarding the signal cascade from BCR to cell cycle entry, and do not show more directed experiments based on such knowledge.

Some additional points are suggested to improve the manuscript:

1. Explain the rationale for looking at NFkB associations better. For the non-experts such an explanation might be helpful.
2. Page 7 sentence "This suggests that AhR expression may have to be maximized...". This sentence is not quite clear and should be rephrased or expanded.
3. The calculation of the replication index, expansion index, or % of dividing cells is not included in the M&M section, although it is pivotal to the study. Please amend.
4. Page 14: On the top of the page the authors phrase that the problem is that B cells do not enter into cell cycle, at the end of the page it is called an expansion defect. This is a bit confusing, and may be rephrased for more consistency across the manuscript and data interpretation.
5. The heat map looks nice, but a Table would be more informative. In a table gene function can be added, numerical values of higher/lower expression, and the p-values.
Can a gene ontology analysis or pathway analysis show more? If this was done and proved without useful results, this can be stated.
6. Cyp1a1 is a strong target of activated AHR, however, there are cells which do not induce this gene. This should be discussed as a caveat somewhere in order not to over-interpret the data.
7. Figure 2 DMSO induces high eYFP, this must be discussed a bit more.
8. As the lentiviral transduction experiments could not be repeated, they should be removed together with their implications in the text.
9. Discussion: The presence of a DRE in the IgM 3' enhancer is not mentioned anywhere, albeit it would be relevant for the study.
10. Effects of AHR-deficiency on cell cycle was shown recently for skin (Frauenstein et al, 2013). These and other references could be mentioned as well, beyond p27kip. Are there DREs in the ccno gene?

Minor points

11. State which mice were bred uniquely for this study, and which would become available to the scientific community
12. Page 3 - acknowledge that there are three AHR-deficient strains.
13. Please explain the absolute difference in expression levels in AhR expression of B cells between Figure 1 and EV1a. Do you think this is just experimental variation or indicative of a biological process?
14. Briefly explain mb1 mice when you first talk about them.
15. Add size markers in the Western Blots.
16. Add ChiP method in M&M

1st Editorial Decision

15 August 2016

Thanks for sending me the point-by-point response. I have now had a chance to take a careful look at it.

I appreciate the proposed outline and find that you address the concerns raised in a good way.

Given this I would like to invite a revised version. You can use the link below to upload the revised version.

When preparing your letter of response to the referees' comments, please bear in mind that this will form part of the Review Process File, and will therefore be available online to the community. For more details on our Transparent Editorial Process, please visit our website: http://emboj.embopress.org/about#Transparent_Process

Thank you for the opportunity to consider your work for publication. I look forward to your revision.

Referee #1

The manuscript of Villa et al. studies mice with the B cell specific deletion of the gene that codes for the ligand-activated transcription factor aryl hydrocarbon receptor (AhR). The AhR gene is broadly albeit rather weakly expressed in B cells including CD5 positive B1 cells and plasma cells as has been shown previously (Sherr and Monti 2013). The authors then confirm previous studies that IL4 can increase the expression of AhR. As a ligand-dependent nuclear transcription factor, AhR is activated by environmental chemicals such as dioxin derivatives. Using a YFP reporter located inside the AhR-regulated gene Cyp1a1 the authors show that stimulation of B cells with IL4 and anti-IgM can increase YFP expression in ex vivo cultured B cells.

In Fig 2B and C we showed that anti-IgM treatment, rather than IL-4, drove substantial up-regulation of *Cyp1a1*, although IL-4 could increase *Ahr* expression (Fig 1B). The reason for this is not clear. The use of concomitant anti-IgM and IL-4 treatments throughout the paper, rather than using anti-IgM alone, was originally done to improve survival rate of *in vitro* cultured B cells.

*The expression of this reporter gene, however, is rather low in mice treated with the AhR ligand 3-methylcholanthrene (3-MC) alone. It is only seen when these mice are immunized at the same time. In Fig. 4 the authors finally study the B-cell specific AhR-deficient mice and show that B cells in these mice expand less well than wildtype B cells when cultured with IL-4 plus anti-IgM. Similar data were also obtained in a competitive transfer experiment and they show that the AhR-deficient mice have fewer short-lived plasma cells. In a comparative transcriptome analysis of AhR-deficient and AhR-sufficient B cells the authors identified a small down-regulation of the cell cycle regulator cyclin O (*Ccno*) and they suggest that the reduced expression could be responsible for this reduced expansion of AhR-deficient B cells in culture and in vivo. However, as they mention at the end of their manuscript, they were unable to rescue this phenotype by *Ccno* overexpression.*

Major points:

The major problem of this manuscript is that one learns little new about the role of AhR in normal B cell development and function and that cellular signaling pathways where AhR plays a role remain ill-defined. Specifically, as AhR is a ligand-dependent receptor, one has to assume that, without the exposure to toxins, there must be a natural ligand, but the nature of this ligand remains obscure.

In Fig 2 we showed that BCR-driven activation (anti-IgM) allowed *Cyp1a1* transcription when B cells were concomitantly exposed to the natural endogenous ligand 6-formylindolo[3,2-*b*]carbazole (FICZ - tryptophan derivative). FICZ was shown to be a high affinity physiological AhR agonist, whose metabolites could be found also in human urine samples (Wincent E, 2009).

In the *in vivo* setting (Fig 3), we used the xenobiotic AhR agonist 3-methylcholanthrene (3-MC) to prove that the AhR pathway could be engaged in B cells, when BCR-driven B cell activation increased AhR availability. As compared to FICZ, 3-MC is more stable and slowly degraded by *Cyp1a1* and allowed us to overcome the shortcomings of the *in vivo* labile nature of FICZ. As *Cyp1a1*-driven Cre recombinase is expressed in heterozygote fashion in the *Cyp1a1*-reporter mouse strain, one wild type copy of *Cyp1a1* is available to degrade FICZ and restrain AhR signalling. Thus, FICZ and other tryptophan metabolites are highly likely endogenous ligands that are nevertheless rapidly metabolized, which limits the efficiency of eYFP reporting.

*Here, it would be important to clarify whether AhR indeed needs ligand binding or whether it can also work without a ligand. Based on data shown in Fig. 2B, the second option is rather unlikely. Furthermore, the results presented in Fig.3B suggest an absence of a natural ligand in vivo, considering that the expression of the *Cyp1a1* reporter is not induced unless the mice are treated with an external *Ahr* ligand.*

The endogenous ligand FICZ originates from the UV or visible light-mediated degradation of tryptophan (predominantly in the skin), but other tryptophan- and indole-derived ligands have been described. It is therefore likely that availability of endogenous AhR agonists is not limiting *in vivo*, but as outlined above that AhR stimulation is short-lived and not of sufficiently long duration to allow high enough induction of Cre recombinase to activate the reporter with high efficiency.

Despite these caveats, we used the *Cyp1a1*-reporter system since it allowed us to define AhR pathway activation *in vivo* at the single cell level in a model of antigen-dependent B cell activation.

Identifying a natural ligand or providing proof that Ahr is activated in B cells in vivo without the introduction of an external Ahr ligand is crucial for this study given that the aim of this manuscript is to analyze "the impact of Ahr deficiency on B cell function in the absence of xenobiotic influences" Another problem of this manuscript is that the claim made in the title of the manuscript (optimal B cell proliferation) is not fully explored. The reduced expansion of AhR-deficient B cells observed in vivo and in vitro could be also due to increased apoptosis. That is not excluded currently in the manuscript.

We thank Referee #1 for the useful comment. We have explored the survival of AhR sufficient and deficient B cells after 72h treatment with medium alone or IL-4 (20 ng/ml). These conditions did not induce B cell proliferation and were therefore optimal to assess B cell survival without the confounding factor of proliferation. As shown in Appendix Fig S2A and B, AhR sufficient and deficient B cells showed similar survival rates upon the above mentioned treatments. We further tested whether AhR deficiency may drive apoptosis in B cells by staining for the early apoptotic marker annexin-V. Upon activation of B cells with different concentrations of α -IgM, AhR deficient cells did not show an enhanced propensity to undergo apoptosis (Appendix Fig S2C). We therefore concluded that AhR deficiency did not affect apoptosis of B cells and the reduced expansion of *Ahr*^{-/-} cells, as compared to *Ahr*^{+/+} controls, was caused by reduced proliferation potential.

Furthermore, and more importantly, the mechanism how AhR regulates proliferation remains ill defined.

Specific comment:

In Fig. 1B the authors confirm previously published data that exposure of B cells to IL-4 increases AhR expression and the same is true with anti-IgM treatment. As in the later experiments they often use a combination between anti-IgM and IL-4, it would be important to show here whether this treatment also has a synergetic effect on the expression of the AhR gene. Furthermore, it would be more appropriate not only to show transcription, but also protein expression data in this study.

We agree with the reviewer and have now performed the requested additional experiment. The results are shown in the main body of the paper as Fig 1C-E. Splenic B cells were stimulated with anti-IgM and/or IL-4. As suggested by Referee #1, concomitant stimulation with anti-IgM and IL-4 substantially enhanced *Ahr* expression as compared to the single treatments. Similar results were obtained both at the mRNA and protein level, also corroborating the data in Fig 1B. We may conclude that *in vitro* treatment of B cells with both anti-IgM and IL-4, as used throughout the paper, not only improved B cell survival but also allowed us to better dissect the effects of AhR deficiency in B cells.

This manuscript completely lacks a phenotypic analysis of the B cell compartment in the B cell specific AhR knockout mice. Is the development of these B cells in the bone marrow and in the periphery completely normal? If not, and if there is an altered B cell compartment for example in the spleen, then this could also explain the differences in the expansion of these cells in ex vivo cultures or in vivo transfer experiments. Thus, these data should at least be mentioned if not shown in the supplement by the authors. It would also be important to know whether the phenotype of this mouse is in any way different to the mice with a complete AhR knockout which have been published previously.

We carefully analysed the B cell compartments in both complete AhR deficient mice and B cell-specific AhR deficient mice. B cell subset distribution was similar between the two mouse strains; however steady-state serum immunoglobulin levels were partly affected in complete AhR deficient mice, whereas unaltered in B cell-specific AhR knockout mice. Due to the well-known deficiencies in the mucosal immune system of complete AhR deficient mice (Kiss EA, 2011; Lee JS, 2012; Qiu J, 2012; Li Y, 2011), we wanted to avoid non-B cell intrinsic deficiencies and therefore decided to focus on mice that lacked AhR only in B cells. Two figures have now been added to describe the B cell compartment in complete AhR deficient (Appendix Fig S1) and B cell-specific AhR deficient mice (Fig EV3).

In Fig.2C it would be helpful to include AhR deficient B cells to prove that the expression of the Cyp1a1 reporter is truly AhR dependent.

We showed in Fig 2C (lower row) that induction of eYFP in the reporter mouse is fully inhibited by the AhR antagonist CH223191, which emphasizes the AhR dependency of the reporter induction. As the reporter strain was generated by knocking-in a Cre recombinase construct into the *Cyp1a1* gene, Cre recombinase and subsequent eYFP transcription mirror *Cyp1a1* transcription.

In Fig.5 the authors find AhR deficient B cells to be outcompeted by wild type B cells when transferred into mice, which is attributed to impaired proliferation of AhR deficient B cells. However in 5A the phenotype is only observed in the population of mature B cells. Homeostatic proliferation usually does not significantly contribute to the maintenance of the mature B cell pool. Thus additional experiments addressing the proliferation, survival and maturation of AhR deficient B cells in vivo would be needed to help explain the observed phenotype.

We agree with Referee #1 that homeostatic proliferation, driven in an antigen-independent fashion (Cabatingan MS, 2002), should not contribute to sustaining the pool of mature B cells. We believe that the results in Fig 5A and B indicate that AhR deficiency has an impact in the expansion of B cells upon BCR engagement. AhR is likely not involved in the mechanisms controlling homeostatic proliferation, since no difference was found in the reconstitution potential of *Ahr*^{-/-} and *Ahr*^{+/+} cells in the Pre B and Immature B cell compartments in the bone marrow (Fig 5A). However AhR deficiency affected the mature B cell pool that is shaped by the proliferation of B cells in response to antigens, provided in the bone marrow chimera setting by exposure to the commensal microbiota during the reconstitution period.

Fig.6 Why were non immunized mice used for this experiment? Is the decrease in splenic PCs also observed after a thymus dependent or thymus independent immunization? What is the marginal zone phenotype of these mice? Are the reduced PC numbers a result of decreased marginal zone B cell numbers or, as stated in the manuscript, a result of AhR having an "impact on B cell proliferation"?

We thank Referee #1 for the insightful comment. No differences were recorded in the marginal zone B cell compartment between B cell-specific AhR deficient and sufficient mice (Fig EV3A). To corroborate the results in Fig 6 (now Appendix Fig S3), we challenged *Ahr*^{fl/-} *mb1*^{Cre+} and *Ahr*^{fl/+} *mb1*^{Cre+} mice with the thymus-dependent model antigen NP-CGG. 7 days post-immunization we assessed splenic plasma cells response and found that spleens of B cell-specific AhR deficient and sufficient mice were equally populated by antigen-specific plasma cells (Appendix Fig S3E). We do not have a conclusive explanation for the difference in the splenic plasma cell response between steady-state and NP-CGG-challenged mice and have therefore removed figure 6 from the main figures and placed it as Appendix Fig S3.

In Fig. 7 the authors study the expression of the AhR target gene Ccno in AhR-sufficient and AhR-deficient B cells stimulated with anti-IgM. Is this expression increased by anti-IgM + IL-4 treatment and in the presence of the toxin ligand?

We thank Referee #1 for the useful comment. We have now tested expression of *Ccno* upon concomitant stimulation of B cells with anti-IgM and IL-4 and we found that, as for *Ahr* expression, co-stimulation of B cells with anti-IgM and IL-4 induced substantially more *Ccno* as compared to the single treatments (Fig 6D). This may reflect the increased availability of AhR upon concomitant anti-IgM and IL-4 treatments that resulted in an elevated potential to drive AhR target genes. Similar to *Cyp1a1* induction, we found that B cell stimulation in presence of AhR ligands such as FICZ and 3-MC boosted *Ccno* expression, suggesting that exogenous supplementation of AhR ligands can further promote AhR transcriptional activity (Fig 6E).

Referee #2:

In this manuscript the authors explore the function of the aryl hydrocarbon receptor (AhR) in B cells. The receptor, a ligand-dependent transcription factor, was initially characterized as a sensor for chemical pollutants and to date the physiological ligand in vivo is not known.

Although AhR has been primarily considered a receptor for xenobiotics such as dioxin, in recent years endogenous physiological agonists of AhR have been identified. 6-formylindolo[3,2-*b*]carbazole (FICZ) is a tryptophan derivative generated upon exposure to UV or visible light. FICZ has high affinity for AhR and was found in cells, rodents and humans (Fritsche E, 2007; Wincent E, 2009). Besides FICZ, several other endogenous molecules have been shown to bind and activate AhR (Denison MS and Nagy SR, 2003; Kleman MI, 1994).

The authors present convincing data using a combination of AhR-deficient mice, AhR-reporter mice and chimeric mice that an AhR-deficiency results in reduced antigen-driven B cell proliferation, although the results are not dramatic, and suggest cyclin O, a target of AhR, contributes to the deficiency. These are novel findings that contribute to an understanding of the impact of an inability to sense AhR ligands on B cell responses to antigen.

We thank Referee #2 for the positive comment.

What I found most interesting about the study was the effect of BCR ligation and IL-4 treatment on the response to AhR agonist. What we learn is that BCR crosslinking and IL-4 treatment increase the transcription of Ahr. However, the authors don't provide any insight as to the repercussion of increased levels of Ahr transcripts. For example, do BCR crosslinking or IL-4 treatment result in increased sensitivity to AhR agonist?

BCR crosslinking induces up-regulation of *Ahr* and *Cyp1a1*, when an AhR agonist is present. Although IL-4 induced *Ahr*, it failed to drive *Cyp1a1* transcription at levels comparable to the anti-IgM treatment (Fig 2B and C). We currently do not know the reason for this. The hypothesis proposed by Referee #2 regarding the increased sensitivity to AhR agonists driven by the increased AhR expression is fascinating and we incorporated it in the discussion. Since AhR agonists are not limiting *in vivo* because of the presence of tryptophan by-products such as FICZ, the control of AhR at transcriptional level could represent a strategy to modulate AhR pathway activation.

BCR crosslinking does not appear to induce AhR agonists as in the absence of an exogenous agonist AhR is not translocated to the nucleus and does not induce Cyp1a1 expression. However, there appears to be a peculiar phenomenon that suggest that B cells may be communicating with each other, namely that the fold increases in Ahr (Fig. 1A,B) and Cyp1a1 (Fig. 2B,D) are significantly greater when total CD19+ cells are analyzed versus individual B cell subpopulations.

In Fig 1B and C (now figure 1F) the differences in *Ahr* expression level are due to the different time points rather than to assessment of total CD19⁺ cells vs individual subsets. In Fig 1B *Ahr* expression was tested at 4h post-stimulation, whereas in Fig 1C (now 1F) it was tested 24h post-activation. To facilitate the interpretation of the data, we added panel G to figure 1 that shows *Ahr* expression kinetics upon B cell activation with anti-IgM and IL-4. *Ahr* expression peaked 4h post-challenge and steadily decreased over time.

Cyp1a1 expression in Fig 2B and D was assessed 24h after stimulation. We believe that these differences are due to experimental variability.

To clarify the points raised by Referee #2, we assessed the ability of total CD19⁺ cells vs isolated B cell subsets to induce *Ahr* and *Cyp1a1*, respectively at 4h and 24h post anti-IgM stimulation. As shown in Fig 1F, isolated follicular B cells (FoB) and marginal zone B cells (MZB) have the same potential of inducing *Ahr* expression as total CD19⁺ cells, 4h post activation with anti-IgM and IL-4. The same applies for *Cyp1a1* expression, measured 24h post B cell activation, as shown in Fig 2D.

As the authors point out the B cells appear to respond to tryptophan products in the culture media. Is it possible that B cells produce metabolic products upon BCR crosslinking that alter the response of neighboring cells?

We tested this interesting possibility in a transwell culture experiments using RPMI medium which contains less tryptophan than our standard medium and was previously shown to not cause AhR activation in Th17 cells. This would maximize a potential contribution of an AhR ligand by B cells. As shown in the figure below, we cultured B cells in transwells to test their potential ability upon BCR crosslinking to produce soluble metabolites that may alter the AhR pathway activation in neighboring B cells.

Presence of anti-IgM and IL-4-activated producer B cells (upper chamber) did not have any positive influence on AhR pathway activation (read as *Cyp1a1* induction) in responder B cells (lower chamber), as compared to responder B cells cultivated in absence of producer B cells. This suggests that B cells are not able upon BCR crosslinking to produce metabolites that alter AhR pathway activation in neighboring cells.

Referee figure - B cells do not produce metabolites able to drive *Cyp1a1* expression in neighbour cells.

A Responder CD19⁺ cells were seeded in the lower chamber of a 0.4 µm-pore transwell. The upper chamber was seeded with or without producer CD19⁺ cells. Cells were cultured in RPMI 5% FCS in presence of 10 µg/ml α-IgM + 20 ng/ml IL-4, for 24h. FICZ was used at 250 nM. Cells were activated to induce *Ahr* and sensitize them to eventual AhR ligands produced by producer B cells. RPMI was used to minimize the background *Cyp1a1* induction due to AhR ligands present in IMDM. FICZ was used as positive control.
B qPCR analysis of *Cyp1a1* expression in responder CD19⁺ cells collected from the lower chamber, as indicated in A. *Cyp1a1* expression was normalized to *Hprt1*.

Referee #3

In this paper by Villa et al., the authors study the role of the aryl hydrocarbon receptor (AHR) in murine B cell maturation, proliferation, and affinity maturation. They first demonstrate constitutive and inducible expression of AHR and its activation by agonists, using cyp1a1 induction as a read-out. They then perform a series of in vivo studies, using several highly sophisticated mouse models of conditional B-cell specific AhR-deficiency in combination with a cyp-reporting system. These experiments show that AhR presence or absence results changes in B cell proliferation upon antigenic challenge. Following up on this, they try to identify the responsible factors by gene expression profiling of AhR-deficient B cells. They report a strong down-regulation of cyclin O, and link this to the observed low proliferation of B cells.

Overall this is a complex and well-performed study, which addresses an up-to-date topic. While research on the role of AhR in the immune system has focused in particular on T cells and innate immune cells, there are only few studies on B cells. Moreover, many of these studies deal with B cell lines, not primary B cells.

Nonetheless, the manuscript suffers somewhat from over-interpretation, especially regarding the mechanistic part of identifying the down-stream events regarding proliferation. This part appears still a bit immature, and indeed the authors say that they could not repeat one experiment up to now. This experiment should thus not be used to incept conclusions.

While the final experiment with unbiased RNA sequencing to identify relevant genes is useful, it is surprising that the authors do not connect their findings/lack of their finding to knowledge regarding the signal cascade from BCR to cell cycle entry, and do not show more directed experiments based on such knowledge.

We performed gene ontology analysis on both the lists of up-regulated and down-regulated genes upon AhR deficiency by using the web-based tool ToppGene (www.toppgene.cchmc.org). This tool provided us with a list of gene ontology terms. We then screened the gene ontology list to remove redundant terms using the web-based tool Revigo (www.revigo.irb.hr). We did not identify candidates in the BCR signaling cascade or in cell cycle that would obviously explain the proliferation defect of AhR deficient B cells we identified. We performed the same analysis using the Ingenuity Pathway Analysis tool but similarly did not identify pathways that we could link with our results.

However using ToppGene we identified among the down-regulated genes list the gene ontology term “phosphatidylinositol 3-kinase regulator activity”. This is the only putative link to the BCR signaling cascade we could identify.

Some additional points are suggested to improve the manuscript:

1. Explain the rationale for looking at NFkB associations better. For the non-experts such an explanation might be helpful.

We have rephrased the sentence citing the paper by Vogel et al., which described putative control of AhR expression by NF-κB in fibroblasts, to clarify why we tested this eventuality in B cells.

2. Page 7 sentence "This suggests that AhR expression may have to be maximized...". This sentence is not quite clear and should be rephrased or expanded.

The sentence was rephrased to clarify our hypothesis that BCR engagement, by increasing *Ahr* expression, positively regulates AhR pathway sensitivity to ligands.

3. The calculation of the replication index, expansion index, or % of dividing cells is not included in the M&M section, although it is pivotal to the study. Please amend.

We have now included in the material and methods section the flow cytometry platform used to make the calculations. We also added the description of the parameters % of divided cells, expansion index and replication index.

4. Page 14: On the top of the page the authors phrase that the problem is that B cells do not enter into cell cycle, at the end of the page it is called an expansion defect. This is a bit confusing, and may be rephrased for more consistency across the manuscript and data interpretation.

Our interpretation of the data is that the reduced ability of *Ahr*^{-/-} B cells to enter the cell cycle (progressing from G₀/G₁ to S phase) has an impact in their expansion potential.

AhR deficient B cells are, however, not intrinsically compromised in their proliferation potential, since the replication index showed that *Ahr*^{-/-} B cells could divide as much as *Ahr*^{+/+} counterparts, once they started to divide. AhR deletion rather impacts their ability to undergo cell division, as shown by the decreased fraction of *Ahr*^{-/-} cells progressing to the S phase of the cell cycle.

We have now rephrased all the sentences throughout the paper to clarify the point raised by Referee #3.

5. The heat map looks nice, but a Table would be more informative. In a table gene function can be added, numerical values of higher/lower expression, and the p-values. Can a gene ontology analysis or pathway analysis show more? If this was done and proved without useful results, this can be stated.

As suggested by Referee #3 we performed gene ontology and pathway analysis, but they did not yield insightful results. We have now mentioned this in the results section.

We have replaced the heat map in Fig 7 (now Fig 6) with two more informative tables, one for genes that were down-regulated (Fig 6A) and one for genes that were up-regulated (Fig 6B) in B cells upon AhR deletion. The tables shown in Fig 6 contain: 1. Gene symbols according to the Mouse Genome Informatics database; 2. Average read counts for *Ahr*^{+/+} and *Ahr*^{-/-} samples, helping the reader to quantify the expression level of a given gene; 3. Fold change in expression between *Ahr*^{+/+} and *Ahr*^{-/-} samples; 4. Adjusted *p* value.

We also added in the appendix section the same two tables (Appendix table S1 and S2) showing in addition to the previously mentioned information: 1. ENSEMBL gene ID showing the gene annotation from the ENSEMBL genome database; 2. Full name of the gene according to the ENSEMBL gene database; 3. Brief description of the gene function or biological process in which a given gene has been described.

We did not add any gene ontology terms in the tables since they would have made up a fairly “dry” and not much meaningful list of gene functions.

6. Cyp1a1 is a strong target of activated AHR, however, there are cells which do not induce this gene. This should be discussed as a caveat somewhere in order not to over-interpret the data.

We agree with Referee #3 that some cells may not induce *Cyp1a1* expression upon AhR pathway engagement. However, the *in vitro* data presented in our paper were generated from pure cultures of B cells, which are able to induce *Cyp1a1*. The *in vivo* quantification of *Cyp1a1* expression in Fig 3 (measured as eYFP) was performed with concomitant surface staining of CD19, a highly specific B cell marker. The analysis of *Cyp1a1* induction was thus limited to B cells only, both *in vitro* and *in vivo*.

We are therefore confident that the interpretation of the *Cyp1a1*-related data has not been influenced by any contaminating cell type that may not be able to express the *Cyp1a1* gene.

7. Figure 2 DMSO induces high eYFP, this must be discussed a bit more.

The background levels of eYFP are not due to exposure to DMSO, but are a consequence of encounter with endogenous AhR ligands present in the culture medium (tryptophan derivatives like FICZ), which are able to drive *Cyp1a1* expression even in absence of any deliberate exposure to AhR agonists. It is likely that AhR up-regulation driven by BCR engagement allows AhR activation by the endogenous ligands present in the culture medium. We have previously described the contribution of tissue culture medium-derived AhR ligands in inducing *Cyp1a1* (Veldhoen M, 2009).

8. As the lentiviral transduction experiments could not be repeated, they should be removed together with their implications in the text.

We have removed the lentiviral transduction experiment from the main body of the text and from the expanded view material section.

9. Discussion: The presence of a DRE in the IgM 3' enhancer is not mentioned anywhere, albeit it would be relevant for the study.

It is not obvious to us that the presence of a DRE in the IgM enhancer should have a bearing on the induction of *Ahr* by IgM stimulation. It was this induction rather than any potential later effects of AhR stimulation on IgM expression, which was the focus of these experiments.

10. Effects of AHR-deficiency on cell cycle was shown recently for skin (Frauenstein et al, 2013). These and other references could be mentioned as well, beyond p27kip. Are there DREs in the ccno gene?

We have now added the relevant references in the discussion section and have performed evolutionary conservation analysis to assess the presence of DREs in conserved regions of the *Ccno* gene.

As shown in Appendix Fig S5, the mouse *Ccno* sequence was compared to the human counterpart to highlight conserved regions that are likely to contain evolutionary relevant DREs. Conserved regions are highlighted in red (intergenic regions), blue (exons) or yellow (untranslated regions) and the height of the peaks shows the extent of evolutionary conservation between the sequences. Above the conserved regions we indicated the putative DREs bound by the AhR/ARNT complex, in red are the DREs identified in the mouse sequence, in blue the DREs identified in the human sequence. We placed an arrow to indicate the conserved sequence containing the AhR binding site validated by ChIP in Fig 6F.

Minor points

11. State which mice were bred uniquely for this study, and which would become available to the scientific community

All the mouse lines used in this study are commercially available, except *Cyp11a1^{Cre} R26R eYFP* mice (Colin J Henderson, Division of Cancer Research, University of Dundee Ninewells Hospital And Medical School, Dundee DD1 9SY, UK) and *SW_{HEL}* mice (Robert Brink, Garvan Institute of Medical Research, Sydney NSW 2010, Australia).

12. Page 3 - acknowledge that there are three AHR-deficient strains.

We have added relevant references to acknowledge the three *Ahr^{-/-}* mouse lines.

13. Please explain the absolute difference in expression levels in AhR expression of B cells between Figure 1 and EV1a. Do you think this is just experimental variation or indicative of a biological process?

These experiments were performed several months apart therefore we are confident in saying that the differences are due to inter-experimental variability.

14. Briefly explain *mb1* mice when you first talk about them.

We have now added a brief sentence explaining *mb1^{Cre}* mice in the results section before describing the phenotype of *Ahr^{fl/+} mb1^{Cre+}* and *Ahr^{fl/-} mb1^{Cre+}* mice.

15. Add size markers in the Western Blots.

Size markers were added close to the relevant proteins in the western blots shown throughout the paper.

16. Add ChiP method in M&M

The Chromatin Immunoprecipitation method can be found in the materials and methods section under the “Chromatin immunoprecipitation, RNA extraction, cDNA generation and real time RT PCR” paragraph.

Corresponding Author Name: Brigitta Stockinger

Manuscript Number: EMBOJ-2016-95027R